# Building Coding Agents via Entropy-Enhanced Multi-Turn Preference Optimization

## Abstract

Software engineering presents complex, multi-step challenges for Large Language Models (LLMs), requiring reasoning over large codebases and coordinated tool use. The difficulty of these tasks is exemplified by benchmarks like SWEBench, where current LLMs still struggle to resolve real-world issues. A promising approach to enhance performance is test-time scaling (TTS), but its gains are heavily dependent on the diversity of model outputs. While standard alignment methods such as Direct Preference Optimization (DPO) and Kahneman-Tversky Optimization (KTO) are effective at aligning model outputs with human preferences, this process can come at the cost of reduced diversity, limiting the effectiveness of TTS. Additionally, existing preference optimization algorithms are typically designed for single-turn tasks and do not fully address the complexities of multi-turn reasoning and tool integration required for interactive coding agents. To bridge this gap, we introduce EntroPO, an entropy-enhanced framework that adapts existing preference optimization algorithms to the multi-turn, tool-assisted setting. EntroPO augments the preference objective to explicitly preserve policy entropy and generalizes learning to optimize over multi-turn interactions rather than single-turn responses. We validate EntroPO by fine-tuning a diverse suite of models from different families and sizes (up to 106B parameters). To maximize performance gains from TTS, we further propose a hybrid best-trajectory selection scheme combining a learned verifier model with model-free approaches. On the SWEBench leaderboard, our approach establishes new state-of-the-art results among open-weight models. A 30B parameter model trained with EntroPO ranks 1st on SWEBench-Lite and 4th on SWEBench-Verified on the open-weight leaderboard, surpassed only by models with over 10x more parameters(*e.g.,* >350B). These results highlight the importance of preserving diversity for effective test-time scaling and establish EntroPO as a robust method for building powerful, interactive coding agents.

## 1 Introduction

Large Language Models (LLMs) have achieved impressive breadth across language understanding, coding assistance, and planning. Yet, they still struggle on complex, multi-step software engineering (SWE) tasks that demand reasoning over large codebases and coordinated tool use (*e.g.,* search, execution, and patching) (Yang et al., 2024b; Wang et al., 2025; Xia et al., 2024; Antoniades et al., 2024; Zhang et al., 2024). A promising line of work that improves performance is test-time scaling (TTS)—sampling more trajectories, searching deeper, and verifying candidates, which can uncover higher-quality solutions on challenging instances (Snell et al., 2024; Beeching et al.; Yao et al., 2023; Xu et al., 2024a). However, TTS only helps if the model produces sufficiently *diverse* candidates to explore meaningfully different solution modes.

Recent alignment methods, including Reinforcement Learning from Human Feedback (RLHF) (Ouyang et al., 2022; Ziegler et al., 2019; Bai et al., 2022) and Direct Preference Optimization (DPO) (Rafailov et al., 2023), have been observed to inadvertently reduce generation diversity (Kirk et al., 2023; Padmakumar & He, 2023; Kim et al., 2024; O'Mahony et al., 2024; Murthy et al., 2025). This *diversity collapse* limits the returns of TTS: when a model concentrates

probability mass on a narrow set of responses for a given prompt, additional samples become redundant, and deeper search yields diminishing marginal gains. Prior efforts to preserve diversity during fine-tuning typically target single-turn settings (Slocum et al., 2025; Li et al., 2024; Wang et al., 2024; Lanchantin et al., 2025) or adjust decoding temperatures at inference time (Renze, 2024). These approaches do not directly address multi-turn, tool-using workflows, where diversity must be maintained *throughout the trajectory* to encourage exploration of a sequence of tool calls and partial hypotheses.

To address this limitation, we introduce ENTROPO, an entropy-enhanced preference optimization method for multi-turn SWE agents. Our approach explicitly adds an entropy regularization term to the standard preference optimization objective to preserve policy diversity. Crucially, ENTROPO extends this entropy-regularized objective from single-turn to *multi-turn trajectories*. It provides a general framework that adapts preference optimization algorithms like DPO and KTO to the multi-turn, tool-assisted setting, whereas prior work has largely focused on single-turn DPO (Slocum et al., 2025). By optimizing over multi-turn interactions, ENTROPO aligns the learning process with the sequential nature of complex coding tasks, teaching the model to build better reasoning paths. We also theoretically analyze the closed-form optimal policy with our method.

To maximize the performance gain of TTS, we pair ENTROPO with a hybrid best-trajectory selection scheme. We combine (i) a learned verifier model that scores trajectories with (ii) model-free approaches that favor high-quality trajectories (*e.g.,* passing tests, trajectory steps). This hybrid selector improves sampling effectiveness and amplifies the gains from parallel rollouts.

We empirically validate ENTROPO across a diverse suite of models from different families and sizes (up to 106B parameters) on SWEBENCH-VERIFIED (Chowdhury et al., 2024) and SWEBENCH-LITE (Jimenez et al., 2024). Our approach achieves state-of-the-art results among open-weight models, with our 30B model ranking 1st on SWEBENCH-LITE and 4th on SWEBENCH-VERIFIED (surpassed only by models over 350B, which are 10x larger). Across all evaluations, ENTROPO significantly outperforms standard DPO and KTO in the TTS setting, maintaining higher trajectory diversity, which translates into larger performance gains from increased test-time compute. Our results confirm that the entropy-preserving term is critical to avoid diversity collapse, and our hybrid selector is more effective than model-only or model-free-only selection.

Our contributions are threefold:

- We propose ENTROPO, an entropy-enhanced *multi-turn* preference optimization method tailored to tool-using coding agents that preserves policy diversity during fine-tuning.
- We theoretically analyze the closed-form optimal policy for our multi-turn objective.
- We present state-of-the-art results among open-weight models on SWEBENCH-VERIFIED and SWEBENCH-LITE, showing significant performance gains from test-time scaling.

By addressing the critical challenge of preserving diversity in multi-turn agents, our work paves the way for developing more powerful LLM-based tools capable of tackling real-world software engineering tasks. We release the code, models, and datasets used for our work for reproducibility.

## 2 RELATED WORK

**LLM Post-training.** RLHF has become the standard approach for aligning LLMs with human preferences (Ouyang et al., 2022; Ziegler et al., 2019; Bai et al., 2022; Schulman et al., 2017), but the PPO-style online approach is computationally intensive, as it requires numerous interactions with a reward model or live environment to generate samples during training (Xu et al., 2024b; Wei et al., 2025). To reduce the compute cost, *preference learning* methods replace explicit reward modeling and online RL with simpler, reward-free objectives that require far less compute. DPO (Rafailov et al., 2023) and its variants (KTO (Ethayarajh et al., 2024), SimPO (Meng et al., 2024), OrPO (Hong et al., 2024)) have emerged as competitive and simpler alternatives to PPO-based RLHF.

However, most preference-learning fine-tuning focuses on the *single-turn* setting and does not directly model multi-turn, tool-using trajectories. Recent efforts have begun to extend preference optimization beyond single responses. Xiong et al. (2025) proposed M-DPO, which provided a framework for training multi-turn, tool-assisted agents on math tasks. While this established a foundation for learning from trajectory preferences in the multi-turn setting, we observe that the method suffers from *diversity collapse*, particularly in long-context coding tasks. This limitation is critical

because, for complex coding tasks, the ability to explore a vast solution space is essential (Golubev et al., 2025; Gao et al., 2025). Offline preference objectives often reduce policy entropy, which undermines the exploration of the learned policy (Setlur et al., 2025). Empirical analyses have documented reduced output diversity under alignment fine-tuning (Kirk et al., 2023), and recent studies attribute mode collapse in fine-tuning to characteristics of offline objectives and KL constraints (Slocum et al., 2025; Wang et al., 2024). While recent methods have attempted to modify divergences, decouple KL components, or construct diversity-aware preference pairs to better control diversity (Slocum et al., 2025; Wang et al., 2024; Lanchantin et al., 2025), explicit entropy preservation remains underexplored in the multi-turn setting. For example, SPL (Slocum et al., 2025) decouples the KL divergence into separate cross-entropy and entropy terms to allow for separate control of diversity. However, it mainly focuses on single-turn settings with DPO for tasks requiring a short context length, leaving the complex multi-turn settings untouched.

We address these challenges with ENTROPO, an entropy-enhanced preference optimization framework applicable to both DPO and KTO. To our knowledge, we are the first to provide a rigorous mathematical derivation of the entropy-augmented preference learning objective in the multi-turn setting. Empirically, our method achieves strong performance while preserving exploration throughout the sequence of tool calls. This enables the full potential of test-time scaling to realize larger gains for complex coding tasks.

**LLMs for Software Engineering.** Repository-level SWE benchmarks such as the SWEBENCH (Jimenez et al., 2024; Chowdhury et al., 2024) have accelerated progress on automated bug fixing and patch generation. Agentic systems like SWE-agent (Yang et al., 2024a) introduced interfaces for repository navigation and code editing, while alternative pipelines (*e.g.,* Agentless) (Xia et al., 2024) achieved strong results with simpler localize-and-repair stages. General-purpose agent frameworks such as OpenHands (Wang et al., 2025) provide open tooling for agents and show competitive performance on SWEBENCH. Despite diverse implementations, these systems share core components (planning and tool-use) and must reason over large codebases via sequences of tool calls. This creates a critical need to maintain exploration and diversity throughout trajectories to enable more effective solution space exploration. Our work focuses on this gap by aligning models specifically for multi-turn, tool-using SWE tasks while preserving trajectory-level policy entropy, thereby enabling more effective exploration over repositories.

**Test-Time Inference Strategies.** TTS strategies improve performance by sampling more candidates, searching deeper, and verifying outputs (Snell et al., 2024; Beeching et al.; Yao et al., 2023; Xu et al., 2024a). These include Best-of-$N$ sampling with verifier reranking and structured search methods such as Tree-of-Thoughts (Yao et al., 2023), which let models explore alternative reasoning paths and self-evaluate. However, the returns from TTS depend on candidate diversity: when generations collapse to a narrow solution space, additional samples and deeper search provide diminishing gains. Moreover, self-evaluation is not always reliable, which can lead to the incorrect selection of the best trajectory. We address these challenges by pairing ENTROPO with a hybrid best-trajectory selector that combines a learned verifier model with model-free approaches. This hybrid approach improves robustness to verifier errors and better exploits the increased diversity produced by ENTROPO, yielding stronger empirical gains as test-time compute scales.

## 3 PROPOSED TECHNIQUE

We propose ENTROPO, an entropy-enhanced preference optimization framework for multi-turn, tool-using coding agents. As shown in Figure 1, we use an agent that follows the standard SWE workflow to interact with a sandboxed repository environment and receive execution feedback at each turn. For TTS, we launch parallel rollouts to collect a set of trajectories for each issue. These trajectories are then scored by a hybrid selector that combines a model-based verifier with model-free approaches. The top-scoring trajectory is selected to submit as the final patch and is evaluated by the benchmark tests. Our core contributions are a novel *training objective* that preserves trajectory-level diversity and a *hybrid selection* mechanism that effectively exploits this diversity. To do so, we augment the standard preference optimization objective with an explicit entropy regularization term, which directly encourages the policy to maintain a broader distribution over potential solutions. For the agent itself, we build upon a standard scaffold, avoiding the introduction of new tool schemas.

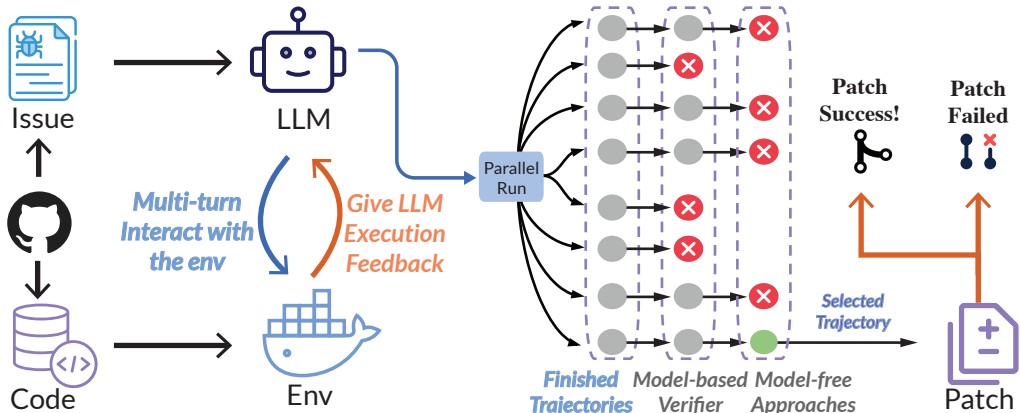

Figure 1: **Overview of** ENTROPO **with TTS.** Given an issue and a repository, an LLM agent interacts with a sandboxed environment over multiple turns, receiving execution feedback. We run parallel rollouts to produce a pool of candidate trajectories. A hybrid selector ranks trajectories using a model-based verifier and model-free approaches, and selects the best trajectory to submit.

## 3.1 PROBLEM SETUP AND ASSUMPTION

We frame the multi-turn, tool-assisted coding task as a finite-horizon episodic Markov Decision Process (MDP), represented by the tuple $\mathcal{M} = \langle \mathcal{S}, \mathcal{A}, H, \mathbb{P}, d_0, u \rangle$. Here, $\mathcal{S}$ is the state space, $\mathcal{A}$ is the action space, $H$ is the maximum number of turns (horizon), $\mathbb{P}$ is the transition dynamics, $d_0$ is the initial state distribution, and $u$ is a trajectory-level utility function.

An initial state $s_1 = x \sim d_0$ corresponds to a coding problem statement. At each step $h \in \{1, \ldots, H\}$, the agent's policy $\pi(a_h|s_h)$ observes the current state $s_h$, which contains the full interaction history, and generates an action $a_h$ (*e.g.,* a bash command). The environment executes $a_h$, returns an observation $o_h$ (*e.g.,* compiler output, test results, or tool feedback), and transitions to the next state $s_{h+1} = (s_h, a_h, o_h)$. This sequential process generates a trajectory $\tau = (x, a_1, o_1, \ldots, a_H, o_H)$, from which we construct a preference dataset $\mathcal{D}$.

Our goal is to optimize the agent's policy $\pi$ using preference feedback, formalized under the following assumption.

**Assumption 3.1.** *We model the probability of preferring one completion over another using the Bradley-Terry model (Bradley & Terry, 1952). Given a problem $x$, the probability that a completion $y^+$ is preferred over $y^-$ ($y^+ \succ y^-$) is given by: $P(y^+ \succ y^-|x) = \sigma(u(x, y^+) - u(x, y^-))$ where $u$ is a latent utility function that scores completions and $\sigma(\cdot)$ is the sigmoid function. The optimal utility function $u^*$ is learned by maximizing the log-likelihood of the observed preferences in a dataset $\mathcal{D}$: $u^* = \arg\max_u \mathbb{E}_{(x,y^+,y^-) \sim \mathcal{D}}[\log \sigma(u(x, y^+) - u(x, y^-))]$.*

In our setting, we rely on **trajectory-level** preference signals. These are supplied by an automated oracle that determines if the final code in a trajectory passes a suite of unit tests. A trajectory that passes is strictly preferred over one that fails.

## 3.2 ALGORITHMIC FORMULATION OF ENTROPO

Standard preference optimization methods, such as DPO (Rafailov et al., 2023) align a policy with a preference dataset by maximizing the likelihood of preferred responses. While effective, this objective often causes the policy to collapse its probability mass onto a narrow set of "winning" solutions. This phenomenon, known as diversity collapse (Murthy et al., 2025), is especially detrimental in complex, multi-step SWE tasks. It hampers the effectiveness of TTS techniques—such as parallel sampling or tree search—because repeated sampling yields redundant candidates, offering diminishing returns and preventing the discovery of potentially superior alternative solutions.

To counteract this, we augment the standard preference optimization objective with a weighted entropy regularization term, $\lambda H(\pi)$. This term directly penalizes low-entropy policies, encouraging

the model to maintain a broader distribution over viable action sequences. This ensures the model not only learns what constitutes a high-quality trajectory but also retains the stochasticity needed to explore a diverse set of candidates at inference time, thereby maximizing the benefit of TTS.

Our full objective, framed as a regularized MDP, is to find the optimal policy $\pi^*$ that maximizes the expected utility with respect to a reference policy $\pi_{ref}$:

$$\max_\pi \mathbb{E}_{x \sim d_0, a_h \sim \pi(\cdot|s_h), o_h \sim \mathbb{P}_h(\cdot|s_h, a_h)}[u(x, y) + \lambda \cdot H(\pi(\cdot|x)) - \beta \cdot D_{KL}(\pi||\pi_{ref})] \qquad (1)$$

where the parameter $\lambda$ promotes diversity and the coefficient $\beta$ penalizes the deviation between the learned policy and the referenced policy $\pi_{ref}$. $H(\pi(\cdot|x)) = -\pi(\cdot|x) \log \pi(\cdot|x)$ denotes the entropy of the learned policy $\pi$. By decoupling the KL divergence, we can rewrite the objective as

$$\max_\pi \mathbb{E}_{x \sim d_0, a_h \sim \pi(\cdot|s_h), o_h \sim \mathbb{P}_h(\cdot|s_h, a_h)}[u(x, y) + \alpha \cdot H(\pi(\cdot|x)) - \beta \cdot H(\pi, \pi_{ref})] \qquad (2)$$

where $\alpha = \lambda + \beta$ and $H(\pi, \pi_{ref}) = -\pi(\cdot|x) \log \pi_{ref}(\cdot|x)$ represents the cross entropy between the learned policy $\pi$ and the referenced policy $\pi_{ref}$.

First, we establish the solution for the single-turn case ($H = 1$), which forms the basis for our multi-turn algorithm.

**Proposition 3.2.** *In the single-turn case ($H = 1$), the optimal policy for the objective in equation 2 is given by $\pi(y|x) \propto \pi_{ref}(y|x)^{\beta/\alpha} \exp(\frac{u(x,y)}{\alpha})$. This optimal policy is identical to the one learned by optimizing the following DPO-style loss function:* $\max_\pi \mathbb{E}_{(x,y^+,y^-) \sim D} \left[ \log \sigma \left( \alpha \left[ \log \frac{\pi(y^+|x)}{\pi_{ref}(y^+|x)^{\beta/\alpha}} - \log \frac{\pi(y^-|x)}{\pi_{ref}(y^-|x)^{\beta/\alpha}} \right] \right) \right]$ *where $y^+ \succ y^-$.*

The proof of Proposition 3.2 can be found in Appendix B. Note that Proposition 3.2 aligns with the conclusion in Slocum et al. (2025), which separates the entropy from the KL divergence term. For the multi-turn case, we follow the derivation framework of Xiong et al. (2025) using backward induction, which is essentially based on Ziebart (2010). The key insight is that the optimal policy and value functions can be defined recursively from the final step $h = H$ to the initial step $h = 1$. This dynamic programming approach leads to the following general solution.

**Proposition 3.3.** *We can recursively define the following Q value functions for a MDP with horizon $H$.*

$$Q_{M,h}(s_h, a_h) = \begin{cases} u(s_h, a_h) & \text{if } h = H \\ E_{o_h \sim \mathbb{P}(\cdot|s_h, a_h)}[V_{M,h+1}(s_{h+1})] & \text{if } h \le H - 1 \end{cases} \qquad (3)$$

*Based on the definition above, we have:*

$$\pi_{\mathcal{M},h}(a_h|s_h) = \frac{\pi_{ref,h}(a_h|s_h)^{\beta/\alpha}}{Z_h(s_h)} \exp(\frac{Q_{\mathcal{M},h}(s_h, a_h)}{\alpha})$$

$$V_{M,h}(s_h) = \mathbb{E}_{a_h \sim \pi_{\mathcal{M},h}(\cdot|s_h)}[Q_{\mathcal{M},h}(s_h, a_h) + \alpha H(\pi(\cdot|s_h)) - \beta H(\pi, \pi_{ref})] = \alpha \log Z_h(s_h) \qquad (4)$$

*where $Z_h(s_h) = \sum_{a_h \in \mathcal{A}} \pi_{ref,h}(a_h|s_h)^{\beta/\alpha} \exp(\frac{Q_{\mathcal{M},h}(s_h,a_h)}{\alpha})$.*

We provide a detailed analysis in Appendix C. This recursive formulation provides the foundation for our ENTROPO training objectives. Additionally, Appendix F offers a more concise, self-contained alternative proof based directly on Ziebart (2010).

ENTROPO- **DPO.** For a preference pair $(\tau^+, \tau^-)$, the loss is:

$$L_{EntroPO-DPO}(\theta)$$
$$= - \sum_{(x, \tau^+, \tau^-) \in D} \log \sigma \left( \alpha \sum_{h=1}^{H} \left[ \log \frac{\pi_{\theta,h}(a_h^+|s_h^+)}{\pi_{ref,h}(a_h^+|s_h^+)^{\beta/\alpha}} - \log \frac{\pi_{\theta,h}(a_h^-|s_h^-)}{\pi_{ref,h}(a_h^-|s_h^-)^{\beta/\alpha}} \right] \right) \qquad (5)$$

ENTROPO- **KTO.** For a dataset $\mathcal{D}$ of trajectories labeled as "desirable" or "undesirable", we first define an implicit reward for a trajectory $\tau$ as $r_\theta(x, y) = \sum_{h=1}^{H} \log \frac{\pi_{\mathcal{M},h}(a_h|s_h)}{\pi_{ref,h}(a_h|s_h)^{\beta/\alpha}}$ The ENTROPO-KTO loss encourages high rewards for desirable trajectories and low rewards for undesirable ones, relative to a margin $z_0$:

$$L_{EntroPO-KTO}(\theta) = E_{x,y \sim D}[\lambda_y - V(x, y)] \qquad (6)$$

where

$$V(x,y) = \begin{cases} \lambda_+ \sigma(\alpha(r_\theta(x,y) - z_0)) & \text{if } \tau \text{ is desirable} \\ \lambda_- \sigma(\alpha(z_0 - r_\theta(x,y))) & \text{if } \tau \text{ is undesirable} \end{cases}$$

Here, $\lambda_+$ and $\lambda_-$ are hyper-parameters weighting the loss for desirable and undesirable examples, respectively, and $z_0 = \mathbb{E}_{x \sim \mathcal{D}, \tau \sim \pi(.|x)} \sum_{h=1}^{H} \left[ -H(\pi(.|s_h)) + \frac{\beta}{\alpha} H(\pi(.|s_h), \pi_{ref}(.|s_h)) \right]$.

### 3.3 TRAINING PIPELINE

Our training process follows a two-stage pipeline. The first stage is SFT, where we teach the base model to use tools reliably. We generate a dataset $\mathcal{D}_{\text{SFT}}$ of successful interaction trajectories using a strong teacher model. The student model is then fine-tuned on these examples to learn stable tool-use patterns of the scaffold.

The second stage is preference learning with ENTROPO. After SFT, we generate a new pool of trajectories by rolling out both the SFT-tuned student model and teacher model. We use a SWE dataset with commit-corresponding test cases for each instance so that we can get the preference label for each trajectory. If the final patch passes the test cases, it is labeled as preferred. Otherwise, it is labeled as not preferred. From this pool, we create a preference dataset $\mathcal{D}_{\text{pref}}$ by pairing trajectories for the same problem, labeling the one with the higher score as preferred. The SFT model is then further fine-tuned on this dataset using our entropy-enhanced objective, which aligns the model with successful problem-solving strategies while preserving the policy diversity crucial for test-time scaling.

### 3.4 TEST-TIME SCALING WITH HYBRID SELECTOR

At inference, we use TTS by running $N$ parallel rollouts for each testing instance, producing trajectories $\{\tau^{(n)}\}_{n=1}^{N}$ from the ENTROPO-tuned policy. We follow prior work (Jain et al., 2025) to apply a hybrid selector that combines model-free approaches with a model-based verifier to prune bad candidates and robustly pick a final patch. The verifier $p_\phi(x, \tau) \in [0, 1]$ is a learned scorer trained on $\mathcal{D}_{\text{pref}}$ with supervised learning. Unlike prior work (Jain et al., 2025) that uses $p_\phi$ as the major ranking criterion, we use it as a conservative filter.

We apply the following filters in order to obtain a candidate set $\{\tau^{(n)}\}$ and then choose a single trajectory:

- **Finished score (model-free).** Discard any $\tau$ truncated by step/token limits: keep only $\mathbb{1}(\text{finished}(\tau) = 1)$.
- **Regression test score (model-free).** Run repository regression checks and keep only trajectories that do not compromise existing functionality: $\mathbb{1}(\text{regress\_free}(\tau) = 1)$.
- **Verifier probability (model-based).** Filter out very unlikely candidates using a low threshold $\eta$: keep $\{\tau \in \mathcal{S} : p_\phi(x, \tau) \geq \eta\}$; we do not rank by $p_\phi$.
- **Step-count heuristic (model-free).** For example, for SWEBENCH-VERIFIED, longer successful trajectories typically reflect broader exploration (*e.g.,* more comprehensive tests) before patch submission. From the remaining set, select $\tau^\star \in \arg\max_{\tau \in \{\tau^{(n)}\}} L(\tau)$, where $L(\tau)$ is the number of environment interactions.

This hybrid selector improves sampling effectiveness and amplifies the gains from parallel rollouts. As $N$ grows, the selector can improve the final solve ratio by pruning failed or low-quality rollouts and favoring well-executed, thoroughly explored solutions.

## 4 EXPERIMENTS

In this section, we show the performance of ENTROPO on benchmarks with R2E (Jain et al., 2025) agent scaffold. §4.1 details datasets, models, and training/inference configurations, including verifier training and the TTS budget. §4.2 presents main results and comparisons to official leaderboard submissions. §4.3 analyzes scaling with the number of parallel rollouts $N$ and ablates ENTROPO components and hyperparameters. Additional implementation details are provided in Appendix E.

Table 1: **Resolve rate (pass@1, %) on benchmarks.** Results are mean $\pm$ std over three runs and the TTS uses $N = 16$ parallel rollouts. For each model, the best result is highlighted in green.

| Model | Origin | SFT | ENTROPO-KTO | ENTROPO-DPO | ENTROPO-KTO+TTS | ENTROPO-DPO+TTS |
|---|---|---|---|---|---|---|
| | | | SWEBENCH-VERIFIED | | | |
| Qwen3-4B | 1.7 ($\pm$ 0.1) | 2.4 ($\pm$ 0.3) | 5.2 ($\pm$ 0.4) | 4.9 ($\pm$ 0.7) | 11.5 ($\pm$ 0.6) | 11.1 ($\pm$ 0.3) |
| Gemma-3-27b | 7.1 ($\pm$ 0.5) | 7.0 ($\pm$ 0.4) | 10.1 ($\pm$ 0.4) | 10.5 ($\pm$ 0.3) | 17.6 ($\pm$ 0.3) | 17.7 ($\pm$ 0.4) |
| Qwen3-Coder-30B | 37.7 ($\pm$ 0.2) | 43.8 ($\pm$ 0.8) | 51.6 ($\pm$ 0.7) | 49.8 ($\pm$ 0.4) | 59.4 ($\pm$ 0.3) | 57.7 ($\pm$ 0.7) |
| GLM-4.5-Air | 51.4 ($\pm$ 0.2) | 51.5 ($\pm$ 0.8) | 53.5 ($\pm$ 0.7) | 52.5 ($\pm$ 0.7) | 58.7 ($\pm$ 0.1) | 57.5 ($\pm$ 0.4) |
| | | | SWEBENCH-LITE | | | |
| Qwen3-4B | 1.2 ($\pm$ 0.3) | 1.3 ($\pm$ 0.5) | 4.8 ($\pm$ 0.4) | 4.7 ($\pm$ 0.5) | 10.0 ($\pm$ 0.8) | 10.4 ($\pm$ 0.4) |
| Gemma-3-27b | 6.0 ($\pm$ 0.8) | 5.9 ($\pm$ 0.4) | 10.4 ($\pm$ 0.2) | 10.4 ($\pm$ 0.7) | 14.6 ($\pm$ 0.6) | 14.4 ($\pm$ 0.7) |
| Qwen3-Coder-30B | 28 ($\pm$ 0.3) | 33.9 ($\pm$ 0.3) | 44.0 ($\pm$ 0.3) | 43.7 ($\pm$ 0.8) | 49.2 ($\pm$ 0.7) | 48.2 ($\pm$ 0.7) |
| GLM-4.5-Air | 43.5 ($\pm$ 0.6) | 43.9 ($\pm$ 0.4) | 44.9 ($\pm$ 0.4) | 44.6 ($\pm$ 0.5) | 48.4 ($\pm$ 0.1) | 47.9 ($\pm$ 0.3) |

## 4.1 IMPLEMENTATION DETAILS

**Datasets.** For SFT tuning, we use the SWE-Smith dataset (Yang et al., 2025), which does not rely on oracles. For preference learning and verifier model training, we use the R2E-Gym-subset (Jain et al., 2025), whose oracles provide trajectory-level utilities. We evaluate on SWEBENCH-VERIFIED and SWEBENCH-LITE, reporting resolve rates based on their official protocols. All performance numbers are pass@1 and no hint or web search is used.

**Models.** We evaluate a diverse set of models from three different families, with sizes ranging from 4B to 106B: Qwen3-4B-Instruct-2507 (Team, 2025), Gemma-3-27b-it (Team et al., 2025), Qwen3-Coder-30B-A3B-Instruct, and GLM-4.5-Air-106B (Zeng et al., 2025a). The verifier is trained with Qwen3-Coder-30B-A3B-Instruct for its strong coding quality and high token throughput.

**Training and Inference.** We train with LLaMAFactory (Zheng et al., 2024b) and set the maximum training sequence length to 18,000 tokens to accommodate long SWE trajectories. To manage memory, we use QLoRA (Dettmers et al., 2023) for GLM-4.5-Air and LoRA (Gao et al., 2021) for the other models. Unless noted, ENTROPO uses $\alpha = 1.1$ and the sensitivity to $\alpha$ is reported in §4.3. During SFT and preference training, we mask system and user prompts and make the LLM response as the learning target. At inference, we allow up to 200 environment interactions per rollout and a maximum sequence length of 131,072 tokens, with temperature 0.7 and top_k $= 20$. For test-time scaling, we run $N = 16$ parallel rollouts for open-weight models. To account for sampling randomness, all experiments on open-weight models are run three times, and we report mean $\pm$ standard deviation.

## 4.2 MAIN RESULTS

**Comparison with Original and SFT-tuned Models.** As shown in Table 1, ENTROPO consistently outperforms both the original and SFT-tuned models across all benchmarks, even without TTS. For models like Gemma-3-27b and GLM-4.5-Air, standard SFT yields minimal gains over the base models. In contrast, ENTROPO delivers substantial improvements, which we attribute to its entropy-regularized objective that preserves policy diversity. This increased diversity is critical for effective exploration and better generalization to unseen problems. When combined with TTS, the performance gains are further amplified, aligning with our theoretical analysis that diversity is key to maximizing the benefits of test-time compute. Note that here the results for Qwen3-Coder-30B are different from the original paper because we use the R2E scaffold instead of the OpenHands scaffold and a much shorter maximum context length due to inference cost consideration.

The impact of ENTROPO is particularly evident for smaller models. For instance, the Qwen3-4B model's performance is negligible after SFT (1.7% on SWEBENCH-VERIFIED and 1.2% on SWEBENCH-LITE), indicating a failure to learn the task. However, with ENTROPO and TTS, its resolve rate surpasses 10%—a remarkable improvement that demonstrates the potential of our approach to make smaller, more efficient models viable for complex SWE tasks.

For larger models, ENTROPO also achieves significant gains. The Qwen3-Coder-30B model trained with ENTROPO-KTO+TTS reaches 59.4% on SWEBENCH-VERIFIED and 49.2% on SWEBENCH-LITE, establishing a strong performance baseline. We note that for GLM-4.5-Air, the improvements from ENTROPO are less pronounced compared to Qwen3-Coder-30B. This is likely due to the use of QLoRA for fine-tuning GLM-4.5-Air, which is not as effective as LoRA.

Table 2: **Resolve rate (pass@1, %) compared to representative** SWEBENCH **leaderboard submissions.** Our entries use the R2E scaffold, and we report the best single run for comparability. Budgets and scaffolds may vary across submissions.

| Submission | Model | Model Size | SWEBENCH-VERIFIED ↓ | SWEBENCH-LITE ↓ |
|---|---|---|---|---|
| | | *Closed Weight Models* | | |
| Refact.ai | Claude3.7/o3/o4-mini | - | 74.4 | 60.0 |
| SWE-agent | Claude 4 Sonnet | - | 66.6 | 56.7 |
| SWE-agent | Claude 3.7 Sonnet | - | 62.4 | 48.0 |
| | | *Open Weight Models* | | |
| OpenHands | Qwen3-Coder | 480B | 69.6 | - |
| OpenHands | Kimi K2 | 1T | 65.4 | - |
| OpenHands | GLM-4.5 | 355B | 64.2 | - |
| ENTROPO-KTO-TTS | Qwen3-Coder | 30B | 59.8 | 49.3 |
| DeepSWE-TTS | Qwen3 | 32B | 58.8 | - |
| ENTROPO-KTO | Qwen3-Coder | 30B | 51.6 | 44.7 |
| Skywork-SWE-TTS | Qwen2.5 | 32B | 47 | - |
| CodeFuse-CGM | Qwen2.5 | 72B | - | 44.0 |
| KGCompass | DeepSeek V3 | 671B | - | 36.7 |
| SWE-fixer | Qwen2.5 | 72B | 24.7 | 32.8 |
| Moatless | Deepseek V3 | 671B | - | 30.7 |

**Comparison to the Official** SWEBENCH **Leaderboard.** In Table 2, we compare our best-performing model against submissions on the official SWEBENCH leaderboard. Our results are highly competitive, particularly among open-weight models. On SWEBENCH-VERIFIED, our 30B parameter ENTROPO-KTO-TTS model achieves a 59.8% resolve rate, surpassed only by models with over 10x more parameters (*e.g.,* >350B). On SWEBENCH-LITE, the same model sets a new state-of-the-art for open-weight models at 49.3%, with our non-TTS version securing the second-highest rank.

Crucially, ENTROPO-KTO-TTS outperforms other TTS-based submissions like DeepSWE-TTS (Luo et al., 2025) and Skywork-SWE-TTS (Zeng et al., 2025b). This highlights the effectiveness of our entropy-preserving training, which preserves the policy diversity essential for maximizing TTS gains. Note that DeepSWE-TTS is online RL-based, showing that our entropy-preserving offline preference learning can be more effective than online RL. Compared to closed-weight models, our results are competitive with top-tier models like Claude 3.7 Sonnet, demonstrating that ENTROPO can significantly narrow the performance gap with commercial models.

### 4.3 ABLATION STUDIES

In this section, we perform ablation studies to investigate the impact of different components of ENTROPO and the sensitivity of the hyperparameters. We mainly focus on the ENTROPO-KTO, as KTO requires fewer GPU memory compared with DPO, as DPO takes a pair of trajectories to calculate the gradient during training.

**Impact of Entropy Regularization.** To isolate the benefit of our entropy-preserving objective, we compare ENTROPO-KTO against the M-KTO (Xiong et al., 2025) and SFT on the Qwen3-Coder-30B model. As shown in Figure 2, ENTROPO-KTO consistently outperforms both baselines. When $N = 1$, ENTROPO-KTO can outperform both SFT and M-KTO, showing its advantage even without TTS. The performance gap between ENTROPO-KTO and multi-turn KTO widens with larger $N$, confirming that explicit diversity preservation is critical for maximizing the gains from TTS. Both preference-based methods outperform SFT, which aligns with findings that SFT can harm generalization (Chu et al., 2025). We conduct a similar experiment on the ENTROPO-DPO model and the M-DPO model, and the results are shown in Figure 5. The results show a similar trend to the ENTROPO-KTO experiment.

**Hybrid Selector Components.** We analyze the contribution of each component of our hybrid selector at $N = 16$. The left plot in Figure 3 shows that removing any single component degrades performance, while the full hybrid selector achieves the best results. This confirms that combining a learned verifier with model-free approaches is the most effective strategy.

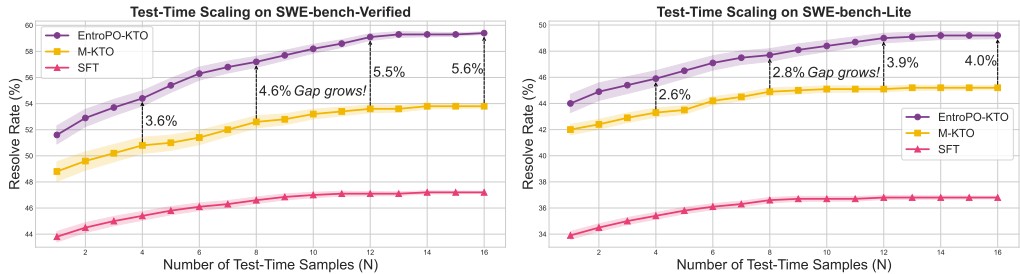

Figure 2: **The Impact of Entropy Regularization on Test-Time Scaling.** Performance of ENTROPO-KTO, M-KTO, and SFT on SWEBENCH-VERIFIED (left) and SWEBENCH-LITE (right) as the number of parallel rollouts ($N$) increases. ENTROPO's entropy regularization consistently yields better scaling.

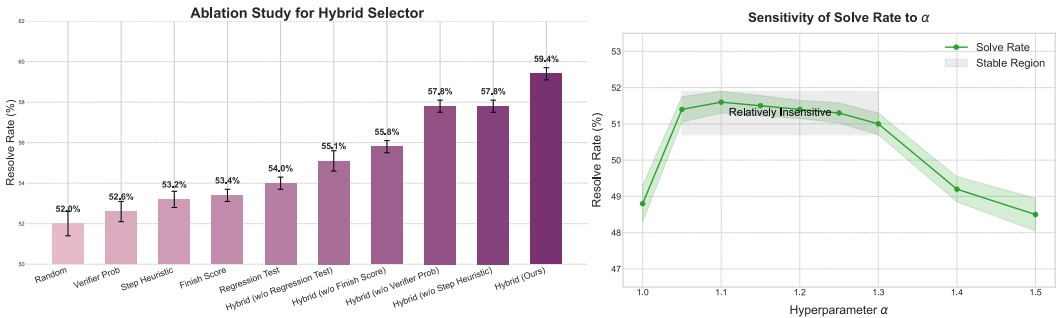

Figure 3: **Ablation Studies on** SWEBENCH-VERIFIED. (Left) Performance contribution of each component in our hybrid selector at $N = 16$. (Right) Sensitivity analysis of the hyperparameter $\alpha$ for ENTROPO-KTO.

**Sensitivity to $\alpha$.** The right plot in Figure 3 shows the performance of ENTROPO-KTO across different values of the hyperparameter $\alpha$. The model is robust to a reasonable range of $\alpha$, indicating that it is not a sensitive hyperparameter. Performance degrades only when $\alpha$ becomes excessively large, causing the training gradients to vanish.

**Impact of Temperature on Performance.** We also investigate whether increasing sampling temperature can replicate the benefits of ENTROPO. As detailed in §E.5, simply raising the temperature fails to deliver comparable performance gains. Instead, it degrades performance by introducing excessive sampling randomness, confirming that temperature tuning is no substitute for the principled entropy regularization of ENTROPO.

## 5 DISCUSSIONS AND LIMITATIONS

While ENTROPO provides a robust framework for enhancing multi-turn agents, we acknowledge several limitations and future directions. Our TTS experiments are limited to $N = 16$ parallel rollouts due to budget constraints. As our results in Figure 2 suggest that performance gains scale with $N$, exploring this behavior with a larger number of rollouts could more conclusively demonstrate the benefits of diversity preservation. Additionally, our end-to-end rollout strategy could be enhanced by incorporating more sophisticated search techniques like Tree-of-Thought (Yao et al., 2023) or solution merging methods (Luong et al., 2025) to explore the solution space more effectively.

Moreover, we acknowledge that the TTS component is an engineering design, guided by empirical observations on SWE tasks, rather than a theoretically guaranteed strategy. As our ablation study in Figure 3 validates, this approach proves to be effective in practice. However, its direct application may not generalize universally to all software tasks or other domains without adaptation. This highlights a promising direction for future research towards the development of a more principled and theoretically backed TTS framework. Such a system might learn an adaptive selection policy

or incorporate uncertainty estimates to move beyond fixed heuristics. We hope our study, which validates the effectiveness of our current empirical design, will inspire the community to explore these more robust and generalizable TTS systems.

Furthermore, due to computational resource constraints, our current implementation of ENTROPO is based on offline preference learning. However, the core principle of entropy regularization can be extended to an online reinforcement learning setting. In such a setup, one could explicitly reward the agent for generating diverse trajectories, potentially leading to even more robust policies. We leave this extension to online RL as a promising direction for future work. Finally, although validated on software engineering, ENTROPO is task-agnostic and could be applied to other complex reasoning domains like competitive mathematics or scientific discovery. We hope our work encourages further exploration into diversity-preserving alignment for building more capable and robust LLM agents.

## 6 CONCLUSION

In this work, we introduce ENTROPO, an entropy-enhanced preference optimization framework designed to improve the performance of multi-turn, tool-using agents on complex SWE tasks. By explicitly regularizing the preference objective to preserve policy diversity, ENTROPO overcomes the limitations of standard alignment methods that often lead to diversity collapse. ENTROPO achieves state-of-the-art results among open-weight models on the SWEBENCH leaderboard, establishing a robust and effective method for building more powerful and reliable coding agents. We hope our work encourages further exploration into diversity-preserving alignment for building more capable and robust LLM agents.

## ETHICS STATEMENT

We have adhered to the ICLR Code of Ethics in conducting this research. Our work focuses on enhancing the capabilities of open-weight LLMs for complex software engineering tasks. By developing methods that improve these publicly accessible models, we aim to foster transparency and reduce the performance gap between open-weight and closed-weight proprietary models. The datasets used for training are derived from publicly available sources, and we are not aware of any personally identifiable information or offensive content within them. Our research does not involve human subjects, and we do not foresee any direct negative societal impacts or ethical concerns arising from our methodology or its outcomes.

## REPRODUCIBILITY STATEMENT

We are committed to ensuring the reproducibility of our research. All implementation details, including datasets, models, and training/inference configurations, are thoroughly described in §4.1. The theoretical analysis of our proposed method is detailed in §3 with complete proofs provided in Appendix B. To facilitate full reproducibility, we will release all our code, the fine-tuned model weights, and the specific data splits used for our experiments upon acceptance of the paper. While the links are withheld for the anonymous review process, we pledge to make all artifacts publicly available on Hugging Face, ensuring that the research community can easily verify, use, and build upon our work.

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

## A The Use of LLMs

In compliance with ICLR 2026 guidelines regarding the disclosure of LLMs usage, we provide the following statement:

Large Language Models are used solely as a general-purpose assist tool for grammar checking and minor stylistic improvements during the writing process of this paper. Specifically, LLMs are employed to:

- Identify and correct grammatical errors in the manuscript
- Suggest improvements to sentence structure and clarity

All research ideas, methodological contributions, experimental results, and substantive written content are entirely the original work of the authors. The authors take full responsibility for all content in this paper, including any text that may have been refined through LLM-assisted grammar checking. The role of LLMs is limited to language polishing and does not rise to the level that would warrant consideration as a contributor to the research.

## B Proof of Proposition 3.2

First, we analyze the optimal policy under the objective

$$\max_{\pi} \mathbb{E}_{y \sim \pi(y|x)}[u(x,y)] + \alpha H(\pi(\cdot|x)) - \beta H(\pi, \pi_{ref})$$

.

By Assumption 3.1, we have

$$u^* = \arg\max_{u} \mathbb{E}_{(x,y^+,y^-)\sim\rho}[\log \sigma(u(x,y^+) - u(x,y^-))] \tag{7}$$

Denote policies $\pi(\cdot|x)$, $\pi_{ref}(\cdot|x)$ and utility function $u(x,\cdot)$ as vectors $\pi$, $\pi_{ref}$, $\mathbf{u}$. We can rewrite our objective as:

$$\max_{\pi} \quad \pi^T\mathbf{u} - \alpha\pi^T\log\pi + \beta\pi^T\log\pi_{ref} = \pi^T(\mathbf{u} - \alpha\log\pi + \beta\log\pi_{ref})$$
$$\text{s.t.} \quad \|\pi\|_1 = 1 \tag{8}$$

The Lagrangian is

$$L(\pi,\lambda) = \pi^T(\mathbf{u} - \alpha\log\pi + \beta\log\pi_{ref}) + \lambda(\sum_i \pi_i - 1) \tag{9}$$

Taking the derivative of the Lagrangian, we have

$$\frac{\partial L}{\partial \pi} = \mathbf{u} - \alpha(\log\pi + \mathbf{1}) + \beta\log\pi_{ref} + \lambda\mathbf{1} = 0 \tag{10}$$

It gives

$$\alpha(\log \pi + \mathbf{1}) = \mathbf{u} + \beta \log \pi_{ref} + \lambda \mathbf{1}$$

$$\log \pi + \mathbf{1} = \frac{\mathbf{u} + \beta \log \pi_{ref} + \lambda \mathbf{1}}{\alpha} \tag{11}$$

$$\pi = \exp^{\frac{\mathbf{u} + \beta \log \pi_{ref} + (\lambda - \alpha)\mathbf{1}}{\alpha}}$$

which implies the optimal policy $\pi^*(y|x) \propto \exp(\frac{u(x,y)}{\alpha})\pi_{ref}(y|x)^{\beta/\alpha}$.

Then, we analyze the optimal policy under the objective

$$\max_{\pi} \mathbb{E}_{(x,y^+,y^-)\sim D} \left[ \log \sigma \left( \alpha \left[ \log \frac{\pi(y^+|x)}{\pi_{ref}(y^+|x)^{\beta/\alpha}} - \log \frac{\pi(y^-|x)}{\pi_{ref}(y^-|x)^{\beta/\alpha}} \right] \right) \right]$$

.

which is equivalent to

$$\max_{\pi} \mathbb{E}_{(x,y^+,y^-)\sim D}[\log \sigma(\alpha \log \frac{\pi(y^+|x)}{\pi(y^-|x)} - \beta \log \frac{\pi_{ref}(y^+|x)}{\pi_{ref}(y^-|x)})]$$

.

Note that $\pi(y|x) = \exp(\frac{1}{\eta}u(x,y))\pi_{ref}(y|x)^{\beta/\alpha}/Z(x)$ is equivalent to $u(x,y) = \alpha \log \pi(y|x) - \beta \log \pi_{ref}(y|x) + \alpha \log Z(x)$. Given the same prompt $x$, compute the utility difference between two different responses $y$ and $y'$. We have

$$
\begin{aligned}
u(x,y) - u(x,y') &= \alpha \log \pi(y|x) - \beta \log \pi_{ref}(y|x) + \alpha \log Z(x) \\
&\quad - [\alpha \log \pi(y'|x) - \beta \log \pi_{ref}(y'|x) + \alpha \log Z(x)] \\
&= \alpha \log \frac{\pi(y|x)}{\pi(y'|x)} - \beta \log \frac{\pi_{ref}(y|x)}{\pi_{ref}(y'|x)}
\end{aligned}
\tag{12}
$$

We now substitute $u(x,y)$ into the Bradley-Terry objective in Assumption 3.1

$$\pi^*(y|x) = \arg\max_{\pi} \mathbb{E}_{(x,y^+,y^-\sim\mathcal{D}}[\log \sigma(\alpha \log \frac{\pi(y^+|x)}{\pi(y^-|x)} - \beta \log \frac{\pi_{ref}(y^+|x)}{\pi_{ref}(y^-|x)})] \tag{13}$$

which satisfies the relationship $\pi^*(y|x) \propto \exp(\frac{u(x,y)}{\alpha})\pi_{ref}(y|x)^{\beta/\alpha}$.

## C  PROOF OF PROPOSITION 3.3

First, we consider the case when $H = 2$. The key idea is to solve the optimization problem through a backward iteration, *i.e.,* from $h = H = 2$ to $h = 1$.

For $h = 2$, by Proposition 3.2, we have:

$$
\begin{aligned}
\pi_{\mathcal{M},2}(\cdot|s_2) &= \arg\max_{\pi_2} \mathbb{E}_{a_2\sim\pi_2(\cdot|s_2)}[u(s_2,a_2) + \alpha H(\pi_2(\cdot|s_2)) - \beta H(\pi_2,\pi_{ref,2})] \\
&\propto \pi_{ref}(\cdot|s_2)^{\beta/\alpha} \exp(\frac{u(s_2,\cdot)}{\alpha})
\end{aligned}
\tag{14}
$$

Then, we define the value function and $Q$ function w.r.t. $\pi_{\mathcal{M},2}$ as:

$$
\begin{aligned}
V_{\mathcal{M},2}(s_2) &= \mathbb{E}_{a_2\sim\pi_2(\cdot|s_2)}[u(s_2,\cdot) + \alpha H(\pi(\cdot|s_2)) - \beta H(\pi,\pi_{ref})] \\
Q_{\mathcal{M},1}(s_1,a_1) &= \mathbb{E}_{o_1\sim\mathbb{P}_1(\cdot|s_1,a_1)}[V_{\mathcal{M},2}(s_2)]
\end{aligned}
\tag{15}
$$

For $h = 1$, we have:

$$
\begin{aligned}
\pi_{\mathcal{M},1}(\cdot|s_1) &= \arg\max_{\pi_1} \mathbb{E}_{a_1\sim\pi_1(\cdot|s_1)}[Q_{\mathcal{M},1}(s_1,a_1) + \alpha H(\pi_1(\cdot|s_1)) - \beta H(\pi_1,\pi_{ref,1})] \\
&\propto \pi_{ref,1}(\cdot|s_1)^{\beta/\alpha} \exp(\frac{u(s_1,\cdot)}{\alpha})
\end{aligned}
\tag{16}
$$

By construction, $\pi_{\mathcal{M},2}$ is already optimal for $h = 2$.

Now, we consider a more general case $h = H$. We could repeat the above process $H$ times starting from $V_{\mathcal{M},H+1} = 0$. We define

$$Q_{M,h}(s_h, a_h) = \begin{cases} u(s_h, a_h) & \text{if } h = H \\ E_{o_h \sim \mathbb{P}(\cdot|s_h,a_h)}[V_{M,h+1}(s_{h+1})] & \text{if } h \leq H - 1 \end{cases} \tag{17}$$

Based on the definition above, we have:

$$\pi_{\mathcal{M},h}(a_h|s_h) = \frac{\pi_{ref,h}(a_h|s_h)^{\beta/\alpha}}{Z_h(s_h)} \exp(\frac{Q_{\mathcal{M},h}(s_h, a_h)}{\alpha})$$

$$V_{\mathcal{M},h}(s_h) = \mathbb{E}_{a_h \sim \pi_{\mathcal{M},h}(\cdot|s_h)}[Q_{\mathcal{M},h}(s_h, a_h) + \alpha H(\pi(\cdot|s_h)) - \beta H(\pi, \pi_{ref})] = \alpha \log Z_h(s_h) \tag{18}$$

where $Z_h(s_h) = \sum_{a_h \in \mathcal{A}} \pi_{ref,h}(a_h|s_h)^{\beta/\alpha} \exp(\frac{Q_{\mathcal{M},h}(s_h,a_h)}{\alpha})$.

To show how the expression of the state value holds, we focus on a general objective:

$$\min_{p \in \Delta(\Omega)} [E_{w \sim p} U(w) - \alpha H(p) + \beta H(p, p_0)] \tag{19}$$

where $\Delta(\Omega)$ denotes the set of probabilities on $w$.

Note that

$$E_{w \sim p} U(w) - \alpha H(p) + \beta H(p, p_0) = E_{w \sim p}[\alpha \log p(w) - \alpha(\frac{\beta}{\alpha} \log p_0(w) - \frac{U(w)}{\alpha})]$$

$$= E_{w \sim p}[\log p(w) - \log(p_0(w)^{\beta/\alpha} \exp(-\frac{U(w)}{\alpha}))] \tag{20}$$

To ensure that $q(w)$ is a valid probability distribution, we define the following normalization constant:

$$Z = \int p_0(w)^{\beta/\alpha} \exp(-\frac{U(w)}{\alpha}) dw \tag{21}$$

Then the normalized distribution is $p^*(w) = q(w)/Z$. Inserting back, we have

$$E_{w \sim p}[U(w)] - \alpha H(p) + \beta H(p, p_0) = \alpha E_{w \sim p}[\log p(w) - \log(Z \cdot p^*(w))]$$

$$= \alpha E_{w \sim p}[\log p(w) - \log p^*(w) - \log Z]$$

$$= \alpha E_{w \sim p} \log \frac{p(w)}{p^*(w)} - \alpha E_{w \sim p}[\log Z] \tag{22}$$

Note that the first item is the KL divergence $KL(p||p^*)$. Thus, the minimizer of the above equation exists when $p(w) = p^*(w)$, *i.e.*, $p^*(w) = \frac{1}{Z} p_0(w)^{\beta/\alpha} \exp(-\frac{U(w)}{\alpha})$. The minimum value is $-\alpha \log Z$. Thus, $V_{\mathcal{M},h}(s_h) = \mathbb{E}_{a_h \sim \pi_{\mathcal{M},h}(\cdot|s_h)}[Q_{\mathcal{M},h}(s_h, a_h) + \alpha H(\pi(\cdot|s_h)) - \beta H(\pi, \pi_{ref})] = \alpha \log Z_h(s_h)$.

By definition, $[\pi_{\mathcal{M},h}]_{h=1}^{H}$ is optimal.

## D    DERIVATION OF ENTROPO LOSS

Based on Proposition 3.3, the optimal policy at step $h$ is

$$\pi_{\mathcal{M},h}(a_h|s_h) = \frac{\pi_{ref,h}(a_h|s_h)^{\beta/\alpha}}{Z_h(s_h)} \exp(\frac{Q_{\mathcal{M},h}(s_h, a_h)}{\alpha}), \tag{23}$$

with normalization factor

$$Z_h(s_h) = \sum_{a_h \in \mathcal{A}} \pi_{ref,h}(a_h \mid s_h)^{\beta/\alpha} \exp(Q_{\mathcal{M},h}(s_h, a_h)/\alpha), \tag{24}$$

and the corresponding value

$$V_{\mathcal{M},h}(s_h) \;=\; \alpha \log Z_h(s_h). \tag{25}$$

Starting from the policy expression,

$$\pi_{\mathcal{M},h}(a_h \mid s_h) \, Z_h(s_h) = \pi_{\text{ref},h}(a_h \mid s_h)^{\beta/\alpha} \, \exp\!\left(\frac{Q_{\mathcal{M},h}(s_h, a_h)}{\alpha}\right)$$
$$\implies \quad \exp\!\left(Q_{\mathcal{M},h}(s_h, a_h)/\alpha\right) = \frac{\pi_{\mathcal{M},h}(a_h \mid s_h) \, Z_h(s_h)}{\pi_{ref,h}(a_h \mid s_h)^{\beta/\alpha}}. \tag{26}$$

Taking the logarithm and multiplying by $\alpha$ gives

$$Q_{\mathcal{M},h}(s_h, a_h) = \alpha \log \pi_{\mathcal{M},h}(a_h \mid s_h) + \alpha \log Z_h(s_h) - \beta \log \pi_{ref,h}(a_h \mid s_h)$$
$$= V_{\mathcal{M},h}(s_h) + \alpha \log \pi_{\mathcal{M},h}(a_h \mid s_h) - \beta \log \pi_{ref,h}(a_h \mid s_h). \tag{27}$$

We can further rewrite Q-value as:

$$Q_{\mathcal{M},h}(s_h, a_h) = \alpha \cdot \log \frac{\pi_{\mathcal{M},h}(a_h|s_h)}{\pi_{ref,h}(a_h|s_h)^{\beta/\alpha}} + V_{\mathcal{M},h}(s_h) \tag{28}$$

With the definition of $Q$-values $Q_{\mathcal{M},h}$, we have

$$\mathbb{E}_{o_h \sim \mathbb{P}_h(\cdot|s_h,a_h)} V_{\mathcal{M},h+1}(s_{h+1}) = \alpha \cdot \log \frac{\pi_{\mathcal{M},h}(a_h|s_h)}{\pi_{ref,h}(a_h|s_h)^{\beta/\alpha}} + V_{\mathcal{M},h}(s_h), \quad \text{if } h \leq H - 1$$
$$u(s_H, a_H) = \alpha \cdot \log \frac{\pi_{\mathcal{M},H}(a_H|s_H)}{\pi_{ref,H}(a_H|s_H)^{\beta/\alpha}} + V_{\mathcal{M},H}(s_H). \tag{29}$$

Summing over $h \in [H]$, we have

$$u(s_H, a_H) = \underbrace{\alpha \sum_{h=1}^{H} \log \frac{\pi_{\mathcal{M},h}(a_h|s_h)}{\pi_{ref,h}(a_h|s_h)^{\beta/\alpha}}}_{(1)} + \underbrace{V_{\mathcal{M},1}(s_1)}_{(2)}$$
$$+ \underbrace{\sum_{h=1}^{H-1} [V_{\mathcal{M},h+1}(s_{h+1}) - E_{o_h \sim \mathbb{P}_h(\cdot|s_h,a_h)} V_{\mathcal{M},h+1}(s_{h+1})]}_{(3)} \tag{30}$$

Term (1) is similar to what we derive in Proposition 3.2. Term (2) can be viewed as a constant when comparing two different responses for the same prompt $s_1$. For Term (3), given the deterministic nature of our tool-integrated coding task, Term (3) is equal to 0. Therefore, we can utilize the maximum likelihood estimation of the utility function with a dataset $\mathcal{D}$ consisting of $(x, \tau^+, \tau^-)$ to obtain our ENTROPO-DPO loss:

$$L_{EntroPO-DPO}(\theta)$$
$$= - \sum_{(x,\tau^+,\tau^-) \in D} \log \sigma \left( \alpha \sum_{h=1}^{H} \left[ \log \frac{\pi_{\theta,h}(a_h^+|s_h^+)}{\pi_{ref,h}(a_h^+|s_h^+)^{\beta/\alpha}} - \log \frac{\pi_{\theta,h}(a_h^-|s_h^-)}{\pi_{ref,h}(a_h^-|s_h^-)^{\beta/\alpha}} \right] \right) \tag{31}$$

With equation 30 implying that Term (3) = 0, the implicit reward $r(x, y)$ is given by $\sum_{h=1}^{H} \log \frac{\pi_{\mathcal{M},h}(a_h|s_h)}{\pi_{ref,h}(a_h|s_h)^{\beta/\alpha}}$. Based on Ethayarajh et al. (2024), we can naturally derive our ENTROPO-KTO loss

$$L_{EntroPO-KTO}(\theta) = E_{x,y \sim D}[\lambda_y - V(x, y)] \tag{32}$$

where

$$r_\theta(x,y) = \sum_{h=1}^{H} \log \frac{\pi_{\mathcal{M},h}(a_h|s_h)}{\pi_{ref,h}(a_h|s_h)^{\beta/\alpha}}$$

$$V(x,y) = \begin{cases} \lambda_+ \sigma(\alpha(r_\theta(x,y) - z_0)) \\ \lambda_- \sigma(\alpha(z_0 - r_\theta(x,y))) \end{cases}$$

$$z_0 = \mathbb{E}_{x\sim\mathcal{D},\tau\sim\pi(.|x)} \sum_{h=1}^{H} \left[ -H(\pi(.|s_h)) + \frac{\beta}{\alpha} H(\pi(.|s_h), \pi_{ref}(.|s_h)) \right]$$

## E  EXPERIMENT DETAILS

In this section, we provide additional details about the experiment details of ENTROPO.

### E.1  DATASET DETAILS

**Training Data.** For SFT, we use the SWE-Smith dataset (Yang et al., 2025), selecting only Python-based instances with problem statements that align with the SWEBENCH format. For preference learning, we employ the R2E-Gym-subset (Jain et al., 2025), which we selected because it contains no repository overlap with our test sets, thereby preventing data leakage.

**Evaluation Data.** We evaluate ENTROPO on the complete official test sets for both SWEBENCH-VERIFIED and SWEBENCH-LITE. Detailed statistics for all datasets are provided in Table 3.

Table 3: **Dataset Statistics**: The table presents the number of training and test samples for each dataset, along with the source of the dataset.

| Dataset | # Train Samples | # Test Samples | Source |
|---|---|---|---|
| SWE-Smith | 8736 | - | https://huggingface.co/datasets/r2e-edits/swesmith-clean |
| R2E-Gym-subset | 4578 | - | https://huggingface.co/datasets/R2E-Gym/R2E-Gym-Subset |
| SWE-bench Verified | - | 500 | https://huggingface.co/datasets/princeton-nlp/SWE-bench_Verified |
| SWE-bench Lite | - | 300 | https://huggingface.co/datasets/princeton-nlp/SWE-bench_Lite |

### E.2  SCAFFOLD DETAILS

In our experiments, we utilize the standard R2E scaffold, which is recognized for its flexibility, ease of use, and robust performance. This scaffold equips the agent with four essential tools: `file_editor` for file editing, `execute_bash` for running bash commands, `search` for file and code retrieval, and `finish` to conclude the task. The complete system prompt, detailing the functionality and parameters of each tool, is provided below.

---

**System Prompt of the Scaffold**

You are a programming agent who is provided a GitHub issue and repository bash environment and is tasked to solve certain tasks (e.g., file localization, testcase generation, code repair, and editing, etc) to resolve the issue.

We have access to the following functions:

— BEGIN FUNCTION #1: `file_editor` —

**Description**: Custom editing tool for viewing, creating, and editing files.

• State is persistent across command calls and discussions with the user
• If path is a file, view displays the result of applying `cat -n`. If path is a directory, view lists of non-hidden files and directories up to 2 levels deep
• The create command cannot be used if the specified path already exists as a file
• If a command generates a long output, it will be truncated and marked with `<response clipped>`
• The `undo_edit` command will revert the last edit made to the file at path

**Notes for using the `str_replace` command**:

---

- The `old_str` parameter should match EXACTLY one or more consecutive lines from the original file. Be mindful of whitespaces!
- If the `old_str` parameter is not unique in the file, the replacement will not be performed. Make sure to include enough context in `old_str` to make it unique
- The `new_str` parameter should contain the edited lines that should replace the `old_str`

**Parameters**:

1. `command` (string, required)
   Allowed values: [`view, create, str_replace, insert, undo_edit`]. The command to run.

2. `path` (string, required)
   Absolute path to file or directory, e.g. `/testbed/file.py` or `/testbed`.

3. `file_text` (string, optional)
   Required for the `create` command. Contains the content of the file to be created.

4. `old_str` (string, optional)
   Required for the `str_replace` command. The exact string in the path to replace.

5. `new_str` (string, optional)
   - Optional for the `str_replace` command to specify the replacement string.
   - Required for the `insert` command to specify the string to insert.

6. `insert_line` (integer, optional)
   Required for the `insert` command. The `new_str` will be inserted after the line number specified here.

7. `view_range` (array, optional)
   - Optional for the `view` command (when path is a file).
   - If provided, specifies the line range to view, e.g. [`11, 12`] shows lines 11 and 12.
   - [`start_line, -1`] will show all lines from `start_line` to the end of file.

8. `concise` (boolean, optional)
   - Optional for the `view` command.
   - Displays a concise skeletal view of the file. If set to `False`, it displays the full content in the specified `view_range`.

— END FUNCTION #1 —
— BEGIN FUNCTION #2: `execute_bash` —
**Description**: Execute a bash command in the terminal.
**Behavior notes**:
- If a command may run indefinitely (long-running), consider running it in the background and redirecting output, e.g. `python3 app.py > server.log 2>&1 &`.
- If the bash command returns exit code -1, it means the process is still running. The assistant may:
  - Call this function again with the command as an empty string ("") to retrieve additional logs.
  - Send more input to STDIN of the running process by calling this function again with the command set to the text input.
  - Send `command="ctrl+c"` to interrupt the currently running process.
- If the command times out, it will be interrupted (SIGINT). The assistant may then retry or do further steps if needed.
**Parameters**:

1. `cmd` (string, required)
   The bash command (and optional arguments) to execute.
   - Can be empty ("") to retrieve more logs if the process is still running.
   - Can be "ctrl+c" to interrupt the running process.

— END FUNCTION #2 —
— BEGIN FUNCTION #3: `search` —
**Description**: Search for a term in a directory or a single file.

- If path is a directory (or unspecified, default is `.` ), it recursively searches all non-hidden files and directories for the search term.
- If path points to a file, it runs a `grep -n` in that file to show line numbers matching the search term.
- If more than 100 files match in a directory search, results are truncated, and the tool will inform you to narrow your search.
- If no matches are found, it will inform you as well.

**Parameters**:

1. `search_term` (string, required)
   The term or string to search for in files.

2. `path` (string, optional)
   The file or directory to search in. Defaults to `.` if not specified.

— END FUNCTION #3 —

— BEGIN FUNCTION #4: `finish` —

**Description**: Finish the interaction once the task is complete or if no further progress can be made.

**Behavior notes**:

- The submit command finalizes your output.

**Parameters**:

1. `command` (string, required)
   Currently allowed value: `[submit]`.

2. `result` (string, optional)
   The result text or final message to submit. Defaults to an empty string if not provided.

— END FUNCTION #4 —

If you choose to call a function ONLY reply in the following format with NO suffix:

```
<function=example_function_name>
<parameter=example_parameter_1>value_1</parameter>
<parameter=example_parameter_2>
This is the value for the second parameter
that can span
multiple lines
</parameter>
</function>
```

`<IMPORTANT>` Reminder:
- Function calls MUST follow the specified format, start with `<function=` and end with `</function>`
- Required parameters MUST be specified
- Only call one function at a time
- VERY IMPORTANT: Each response must include both reasoning (as natural text) and function call (in the above format) to solve the task.

### E.3 RUNNING ENVIRONMENT

Our implementation relies on the LLaMA-Factory (Zheng et al., 2024b) for model training and SGLang (Zheng et al., 2024a) for efficient inference deployment. All experiments are performed on a server configured with four 32-core AMD EPYC 7702 CPUs, 8 NVIDIA H100 (80GB) GPUs, and 4 NVIDIA A100 (40GB) GPUs. The H100 GPUs are dedicated to the primary training and inference workloads, while the A100 GPUs provided supplementary computational support during inference.

### E.4 TRAINING AND INFERENCE DETAILS

**Training Hyperparameters.** We configure our training process as follows. For LoRA, we set `lora_rank=8`, `lora_alpha=16`, and apply it to all available modules (`lora_target=all`).

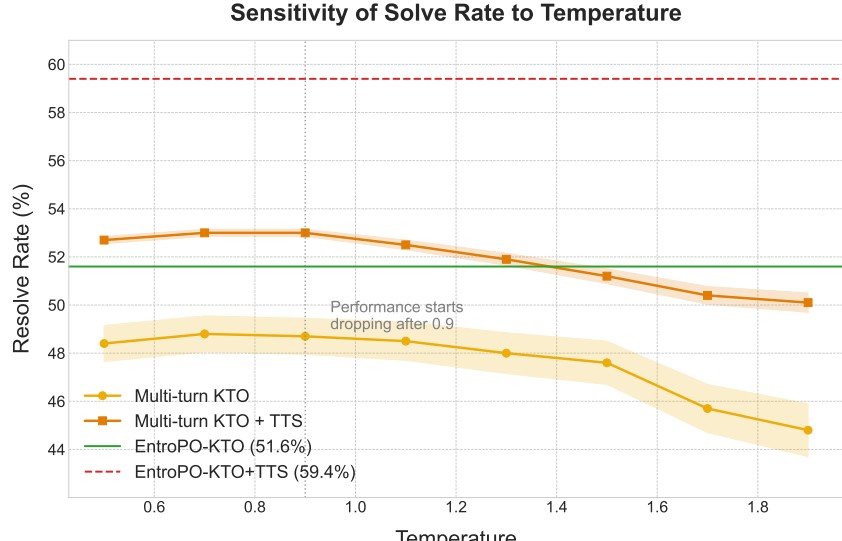

Figure 4: **Impact of Temperature on Performance.** Performance of multi-turn KTO and KTO+TTS on SWEBENCH-VERIFIED with varying temperature, compared to ENTROPO with a fixed temperature of 0.7. Increasing temperature fails to match the performance of ENTROPO and degrades performance past 0.9.

For the GLM-4.5-Air model, we use 4-bit QLoRA quantization. Across all models, we use a context length of 18,000 tokens, a learning rate of 1e-5, and a warmup ratio of 0.1. This context length is chosen to avoid Out of Memory errors during training as a longer context length would require much more memory. The batch size is set to 4 for GLM-4.5-Air and 16 for all other models. Both the SFT and preference optimization stages are trained for a single epoch using these settings.

**Inference Parameters.** During inference, each rollout is permitted a maximum of 200 environment interaction steps and a total generation length of up to 131,072 tokens. We use a sampling temperature of 0.7, `top_k` of 20, and `top_p` of 0.8 for all models, which is the recommended setting from Qwen3 model card.

**Hybrid Trajectory Selection.** Our hybrid selector employs a multi-stage filtering process to identify the best trajectory from the generated candidates. The process is as follows:

1. **Initial Filtering**: We first discard any trajectories that are truncated due to exceeding the step or token limits.

2. **Regression Testing**: Next, we filter out trajectories that fail the regression tests generated by the R2E-Gym framework.

3. **Verifier-Based Filtering**: We then apply a verifier model and remove trajectories with a probability below a threshold of 0.01. We choose a conservative threshold because we empirically observe that many valid solutions do not receive high probability scores, but scores below 0.01 are a strong indicator of a flawed trajectory.

If any filtering step results in an empty set of candidates, we revert to the candidate pool from the previous step.

After filtering, we apply a final model-free heuristic for selection. We select the trajectory with the **most** environment interaction steps for SWEBENCH-VERIFIED and the **fewest** for SWEBENCH-LITE. This distinction is motivated by the nature of the benchmarks and our observation. For SWEBENCH-VERIFIED, every instance has been manually verified by human engineers at OpenAI (Chowdhury et al., 2024) to have accurate problem statements and robust environments. Thus, a longer successful trajectory can be a positive signal, potentially indicating more comprehensive reasoning, the addition of more thorough test cases, or the consideration of more corner cases before submitting a final patch. However, for SWEBENCH-LITE, the situation is different. It is known to contain instances with

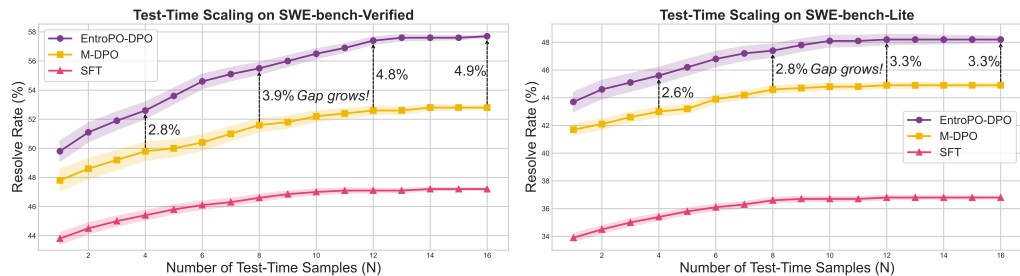

Figure 5: **The Impact of Entropy Regularization on Test-Time Scaling.** Performance of ENTROPO-DPO, M-DPO, and SFT on SWEBENCH-VERIFIED (left) and SWEBENCH-LITE (right) as the number of parallel rollouts ($N$) increases. ENTROPO's entropy regularization consistently yields better scaling.

misleading or incomplete problem statements. Xia et al. (2024) and Chowdhury et al. (2024) note that the original SWE-bench dataset contains underspecified problem statements and problematic environment setups that cause some unit tests to fail regardless of the solution. In such cases, a long trajectory can signal that the agent is misled by a vague prompt or is "hallucinating" complexity in response to incorrect unit test feedback. Therefore, we adopt the strategy from prior work (Agarwal et al., 2025; Hassid et al., 2025) to prefer shorter solutions, which helps mitigate the negative impact of these problematic instances. We conduct ablation studies on the impact of this strategy in §E.5 which shows that this strategy is effective on SWEBENCH-LITE.

### E.5 ADDITIONAL EXPERIMENTS

**Impact of Temperature on Performance.** To investigate whether increased sampling diversity can replicate the benefits of our entropy-regularized approach, we evaluate the multi-turn KTO and multi-turn KTO+TTS models under varying temperatures. We test the Qwen3-Coder-30B model on SWEBENCH-VERIFIED with temperatures ranging from 0.5 to 1.8 and compare its performance to ENTROPO-KTO and ENTROPO-KTO+TTS, which use a fixed temperature of 0.7. As shown in Figure 4, increasing the temperature provides no performance benefit. In fact, performance begins to decline beyond a temperature of 0.9, as excessive sampling randomness undermines the precision required for SWE tasks involving tool use and code editing. These results demonstrate that merely increasing temperature is not a substitute for principled entropy regularization, as it fails to match the performance gains achieved by ENTROPO.

**Impact of Entropy Regularization on DPO.** To investigate whether our entropy-regularized approach can benefit DPO, we conduct experiments on the Qwen3-Coder-30B model with the ENTROPO-DPO and M-DPO (Xiong et al., 2025) models. We test the Qwen3-Coder-30B model on SWEBENCH-VERIFIED and SWEBENCH-LITE with different number of parallel rollouts ($N$) and compare its performance to ENTROPO-DPO, M-DPO, and SFT. As shown in Figure 5, ENTROPO-DPO consistently outperforms M-DPO and SFT. When $N = 1$, ENTROPO-DPO can outperform both SFT and M-DPO, showing its advantage can be independent of TTS. As $N$ increases, the performance gap between ENTROPO-DPO and M-DPO widens, similar to our findings in Figure 2.

**Impact of Step Heuristic on SWEBENCH-LITE.** We adopt different step heuristics for ENTROPO-KTO with model Qwen3-Coder-30B on SWEBENCH-LITE as follows: 1) Prioritize the trajectory with the most environment interaction steps. 2) Prioritize the trajectory with the fewest environment interaction steps. 3) Randomly select a trajectory as the step heuristic. The results are shown in Table 4. We can observe that when using the fewest steps heuristic, the performance of ENTROPO-KTO is best, which verifies our motivation in §E.4. However, even using the longest steps heuristic or random selection heuristic, the performance of ENTROPO-KTO is still better than the best open-source submission on SWEBENCH-LITE (CodeFuse-CGM with 44.0% resolve rate), which demonstrates the robustness of our overall framework.

**Heuristics for a Random Real-World SWE Problem.** The above analysis leads to a practical guide for choosing a heuristic for a random real-world SWE problem:

- **If the developer trust the specification of the SWE problem:** Favor Longest Steps.
- **If the specification is vague/noisy:** Favor Shortest Steps.
- **If the developer has no idea (The Default Strategy):** We recommend favoring Longest Steps.

As shown in Table 4, when we applied the "Longest Steps" heuristic to SWEBENCH-LITE (where it is empirically suboptimal), the performance (47.3%) was comparable to Random Selection (47.1%). This means that even if the heuristic is "suboptimal" for the data distribution, it causes no significant harm. However, on high-quality data (SWEBENCH-VERIFIED), using Longest Steps yields significant gains. Therefore, prioritizing the longest trajectory can be a default choice—it captures the upside on good data without degrading performance on noisy data.

Table 4: **Impact of Step Heuristic on** SWEBENCH-LITE**.**: The table presents the performance of different step heuristics on SWEBENCH-LITE. We compare prioritizing trajectories with the most steps, fewest steps, and random selection.

|  | Most Steps | Fewest Steps | Random |
|---|---|---|---|
| **Resolve Rate (%)** | 47.3 ($\pm$ 0.3) | 49.2 ($\pm$ 0.7) | 47.1 ($\pm$ 0.6) |

## F    ALTERNATIVE DERIVATION OF ENTROPO-DPO

Building on the framework of entropy-regularized MDPs (Ziebart, 2010), we define the regularized value function for any policy $\pi$ as:

$$V^\pi(s) = \mathbb{E}_\pi\left[\sum_{h=1}^{H}\left(u(s_h, a_h) - \alpha \log \frac{\pi(a_h|s_h)}{\pi_{\text{ref}}(a_h|s_h)^{\beta/\alpha}}\right)\Bigg| s_1 = s\right] \tag{33}$$

To facilitate the analysis, we define a modified reward function $r'$ that incorporates the reference policy:

$$r'(s, a) = r(s, a) + \beta \log \pi_{\text{ref}}(a|s) \tag{34}$$

Using this modified reward, the regularized Q-function of a policy $\pi$ is defined as:

$$Q^\pi(s, a) = r'(s, a) + \mathbb{E}_{s' \sim p(\cdot|s,a)}[V^\pi(s')] \tag{35}$$

The relationship between the value function and Q-function is:

$$V^\pi(s) = \mathbb{E}_{a \sim \pi(\cdot|s)}[-\alpha \log \pi(a|s) + Q^\pi(s, a)] \tag{36}$$

The regularized optimal policy $\pi^*$, which maximizes $V^\pi(s)$, is characterized by the optimal Q-function $Q^*$ and value function $V^*$. The optimal policy is given by:

$$\pi^*(a|s) = \exp\left(\frac{Q^*(s, a) - V^*(s)}{\alpha}\right) \tag{37}$$

By substituting this form of $\pi^*$ back into the sum of the log-ratios, we can rewrite the sum of the log-ratios as follows:

$$\sum_{h=1}^{H} \alpha \log \frac{\pi^*(a_h|s_h)}{\pi_{\text{ref}}(a_h|s_h)^{\beta/\alpha}} = \sum_{h=1}^{H} (Q^*(s_h, a_h) - V^*(s_h) - \beta \log \pi_{\text{ref}}(a_h|s_h)) \tag{38}$$

$$= \sum_{h=1}^{H} [u(s_h, a_h) + \beta \log \pi_{\text{ref}}(a_h|s_h) + \mathbb{E}_{s' \sim p(\cdot|s_h, a_h)} V^*(s') - V^*(s_h) \tag{39}$$

$$- \beta \log \pi_{\text{ref}}(a_h|s_h)] \tag{40}$$

$$= \sum_{h=1}^{H} (u(s_h, a_h) - V^*(s_1)) + \sum_{h=1}^{H-1} \left[ \mathbb{E}_{s' \sim p(\cdot|s_h, a_h)}(V^*(s')) - V^*(s_{h+1}) \right] \tag{41}$$

$$\tag{42}$$

Given the deterministic nature of our software engineering task, we have

$$\sum_{h=1}^{H-1} \left[ \mathbb{E}_{s' \sim p(\cdot|s_h, a_h)}(V^*(s')) - V^*(s_{h+1}) \right] = 0 \tag{43}$$

This simplification leads to the final relationship linking the total reward to the regularized policy and the initial value:

$$\sum_{h=1}^{H} u(s_h, a_h) = \sum_{h=1}^{H} \alpha \log \frac{\pi^*(a_h|s_h)}{\pi_{\text{ref}}(a_h|s_h)^{\beta/\alpha}} + V^*(s_1) \tag{44}$$

Finally, combining this result with Assumption 3.1 allows us to derive the EntroPO-PPO loss.

