# OpenReview forum: "Building Coding Agents via Entropy-Enhanced Multi-Turn Preference Optimization"
_ICLR.cc/2026/Conference — Submitted to ICLR 2026_

### Official Review · Reviewer_q16a · 2025-10-25

**Soundness:** 3
**Presentation:** 2
**Contribution:** 2
**Rating:** 4
**Confidence:** 4

**Summary:**

The paper introduces ENTROPO, an entropy-enhanced preference optimization framework for training multi-turn, tool-using coding agents. The key insight is that standard alignment methods like DPO/KTO reduce output diversity, which weakens the benefit of test-time scaling (TTS) in challenging SWE tasks. ENTROPO augments preference optimization with explicit entropy regularization to preserve policy diversity, and generalizes it from single-turn to multi-turn trajectories. The method is paired with a hybrid best-trajectory selector combining a learned verifier with model-free filters. Experiments across multiple model families demonstrate that ENTROPO consistently improves performance with or without TTS.

**Strengths:**

* The paper extends entropy-preserving preference optimization from single-turn to multi-turn, adapting that for agentic coding tasks.
* Empirical results are strong, showing clear improvement over baseline methods.
* There are detailed ablation studies demonstrating the impact of each module in training and in the TTS pipeline.

**Weaknesses:**

* Methodology-wise, this paper is largely an extension of SPL [1] to the multi-turn scenario. The objective in Proposition 3.2 is the same as Equation 5 in [1]. While [1] was cited in related works, I would suggest clearly stating the relationship with that paper at the beginning of Section 3, to highlight the unique contribution of this paper.
* The extension of SPL to the multi-turn scenario is relatively straightforward. One can bypass the derivation in Appendix C by taking a similar approach to Step-DPO, which decomposes a multi-turn trajectory of $H$ turns into $H$ single-turn trajectories, and arrive at the same objective function.
* The introduction of ENTROPO-KTO feels abrupt at the end of Section 3.2, as the preceding discussion was mainly centered around DPO. In addition, is there a reason to conduct all ablations with KTO rather than DPO?
* Using the longest for SWE-verified and the shortest for SWE-Lite in the step-count heuristic in TTS is somewhat overfitting to the test sets.


References

[1] Slocum, Stewart, Asher Parker-Sartori, and Dylan Hadfield-Menell. "Diverse preference learning for capabilities and alignment." The Thirteenth International Conference on Learning Representations. 2025.

[2] Lai, Xin, et al. "Step-dpo: Step-wise preference optimization for long-chain reasoning of llms." arXiv preprint arXiv:2406.18629 (2024).

**Questions:**

* How are the trajectories scored in Sec 3.3? Did you use a binary score or a more fine-grained one?

---

> ### Author Response · Authors · 2025-11-20
> **Response to Reviewer q16a (1/3)**
>
> We thank Reviewer q16a for the detailed and constructive feedback. We detail the response for each concern below.
>
> > Methodology-wise, this paper is largely an extension of SPL [1] to the multi-turn scenario. The objective in **Proposition 3.2** is the same as **Equation 5** in [1]. While [1] was cited in related works, I would suggest clearly stating the relationship with that paper at the beginning of **Section 3**, to highlight the unique contribution of this paper. The extension of SPL to the multi-turn scenario is relatively straightforward. One can bypass the derivation in **Appendix C** by taking a similar approach to Step-DPO, which decomposes a multi-turn trajectory of turns into  single-turn trajectories, and arrive at the same objective function.
>
>
> **Response**
>
> We thank the reviewer for this sharp observation. The reviewer is correct that our single-turn objective in **Proposition 3.2** shares the same conclusion with SPL [1]. We will update **Section 3** to make this relationship explicit.
>
> However, we must respectfully but strongly disagree that our paper is "largely an extension." SPL and our EntroPO have different objective functions. SPL attempts to decouple the standard KL-regularized objective into its cross-entropy and entropy components. Our EntroPO is different: to solve the diversity collapse issue, we intentionally formulate a new objective by adding an explicit, separate entropy regularization term to the reward, in addition to the KL divergence term. This is not a simple re-derivation but a new objective design. Furthermore, our work makes two critical, non-trivial theoretical contributions that are not present in [1].
>
> **1. A Correct Proof for the Single-Turn Case**
>
> Regarding **Proposition 3.2**, we want to clarify a crucial point. The derivation of a similar objective in SPL [1] is incorrect, and **it is impossible to derive our **Proposition 3.2** by following their flawed proof.** The error in [1] stems from an incorrect objective in their Equation (27). Specifically, in **Equation (25)** the authors aim to maximize $ \text{reward} + \alpha \cdot \text{entropy} - \beta \cdot \text{cross-entropy} $, but in the rewritten form (Equation (27), p. 16) they mistakenly place a minus sign in front of $ \beta\,\pi^\top \log \pi_{\text{ref}} $ instead of a plus sign (since $ -\beta \cdot \text{cross-entropy} = + \beta\,\pi^\top \log \pi_{\text{ref}} $). This means the entire SPL derivation is based on an incorrect objective. In contrast, we provide (to the best of our knowledge) the first complete and correct derivation for this single-turn, entropy-enhanced objective. Our rigorous proof, which resolves these issues in [1], is given in **Appendix B** and forms a necessary theoretical foundation for the entire framework.
>
> **2. A Non-Trivial Multi-Turn Derivation: Why a Step-DPO Analogy is Not Applicable**
>
> We thank the reviewer for the suggestion regarding a Step-DPO-like decomposition. However, we must clarify that such an approach is not applicable here due to a fundamental difference in problem structure.
>
> The decomposition in methods like Step-DPO is valid because it operates on preferences over actions that branch from an **identical context**. That is, it compares a winning action ($a_{win}$) versus a losing action ($a_{lose}$) given a shared, common history ($s_0, s_1, ..., s_{k-1}$).
>
> Our framework, by contrast, is designed for a multi-turn, interactive setting. Grounded in the Bradley-Terry model, we learn from preferences over **entire trajectories** ($\tau^+$ vs. $\tau^-$). These trajectories are path-dependent and represent two distinct histories. After the very first step, the states in each trajectory diverge ($s_h^+ ≠ s_h^-$ for $h > 0$). In the general case, the preferred and dispreferred trajectories may have no state-action overlap whatsoever.
>
> Because there is no common context or shared prefix between the two trajectories being compared, the problem cannot be decomposed into a sum of independent, single-turn decisions. Our objective function (Eq. 5) correctly reflects this by summing log-policy ratios along two entirely different state-action paths.
>
> Therefore, the derivation in **Appendix C and D** is a necessary and non-trivial contribution. It provides the first formal theoretical grounding for this entropy-enhanced objective in a truly interactive, multi-turn setting. We will clarify this distinction in the final paper.
>
> In summary, we *fix* the single-turn proof (Appendix B) and provide the *first* multi-turn derivation (Appendix C and D) for this entropy-enhanced objective.

---

> ### Author Response · Authors · 2025-11-20
> **Response to Reviewer q16a (2/3)**
>
> > The introduction of ENTROPO-KTO feels abrupt at the end of Section 3.2, as the preceding discussion was mainly centered around DPO. In addition, is there a reason to conduct all ablations with KTO rather than DPO?
>
> **Response:** We thank the reviewer for this question regarding EntroPO-KTO and our choice to use it for the ablation studies.
>
> We would like to respectfully argue that we have been mentioning the KTO from the abstract and Introduction as an important approach in preference learning. Our intention was to show that EntroPO is a general framework applicable to various preference optimization algorithms, with DPO and KTO being one of the most prominent examples, as we have mentioned in Introduction that EntroPO can be applied to both DPO and KTO.
>
> The primary reason for conducting our main ablation studies with KTO was a practical one: memory efficiency. The DPO loss function operates on pairs of trajectories (a "preferred" and "dis-preferred" example for each data point). In contrast, the KTO loss function processes individual trajectories (labeled as "desirable" or "undesirable"). In our setting, where we work with very long multi-turn trajectories, using KTO allowed us to fit slightly larger and more stable batches into GPU memory. Given that KTO's performance is recognized as competitive with DPO, it served as an excellent and resource-friendly choice for these detailed studies.
>
> To address this feedback and improve the manuscript, we have added a discussion in the ablation study section (**Section 4.3**) to explicitly state our rationale for using KTO.
> Furthermore, to demonstrate that our findings are not specific to KTO, we have added a complementary ablation study using EntroPO-DPO in Appendix E (**Figure 5**). These results confirm that the benefits of our entropy regularization hold for the DPO variant as well, reinforcing the generality of our core claims.
>
> We appreciate this feedback, as it has helped us improve the clarity of our methodology and the robustness of our experimental validation.
>
>
> > Using the longest for SWE-verified and the shortest for SWE-Lite in the step-count heuristic in TTS is somewhat overfitting to the test sets.
>
> We thank the reviewer for raising this insightful and important point. We appreciate the opportunity to elaborate on our choice, which we believe is a matter of engineering design rather than overfitting.
>
> First, we would like to clarify that using trajectory length as a selection heuristic is a recognized approach in the literature on test-time scaling and complex reasoning. Both preferring longer trajectories [2,3] and shorter ones [4,5] are established strategies, with the optimal choice depending on the specific task and dataset characteristics. As noted in [3], the influence of the reasoning step length can vary significantly across different applications.
>
> Our choice was based on a careful analysis of the distinct properties of the two benchmarks.
>
> For SWEBench-Verified, every instance has been verified as solvable by human engineers at OpenAI. In this context, a longer successful trajectory can be a positive signal, potentially indicating more comprehensive reasoning, the addition of more thorough test cases, or the consideration of more corner cases before submitting a final patch.
>
> For SWEBench-Lite, the situation can be different. It is known to contain instances with misleading or incomplete problem statements. This observation is strongly supported by the analysis in [6], which notes that problem descriptions in SWE-Bench-Lite often lack sufficient or accurate information, making them ambiguous or even unsolvable. In such cases, a long trajectory often signals that the agent is engaged in unproductive exploration or is stuck in a loop due to the misleading prompt. Therefore, we adopted the strategy from work [5] to prefer shorter, more direct solutions, which helps mitigate the negative impact of these problematic instances.
>
> Given these differences, we respectfully argue that our approach is not overfitting, but rather an engineering design tailored to the known characteristics of each benchmark.
>
> To further address the reviewer's valid concern, we have conducted new experiments, now included in **Appendix E.5 (Table 3)**, where we apply different step-count heuristics to SWEBench-Lite. The results show that applying the "longest trajectory" heuristic indeed leads to a drop in performance compared to our chosen "shortest" heuristic, empirically validating our design choice. Critically, we also note that even with this suboptimal heuristic, EntroPO performance remains highly competitive, surpassing the best-performing open-weight submission on the leaderboard. This underscores the robustness of our core training method.
>
> We have added this detailed discussion and the new results to the revised **Appendix E.5** to make our rationale and its empirical support fully transparent. We thank the reviewer for prompting this valuable clarification.

---

> ### Author Response · Authors · 2025-11-20
> **Response to Reviewer q16a (3/3)**
>
> > How are the trajectories scored in Sec 3.3? Did you use a binary score or a more fine-grained one?
>
> **Response:** Thanks for asking. We adopt a standard approach to determine the preference for each trajectory during training by using the automated oracle based on commit-corresponding test cases, which is a common approach in SWE tasks [7,8]. Trajectories that can pass all those test cases are labeled as preferred, and those that cannot are labeled as dispreferred. We have added this clarification in the revised version's **Section 3.3** to make it clearer. We thank you for your feedback to help us improve the paper.
>
> References:
>
> [1] Slocum, Stewart, Asher Parker-Sartori, and Dylan Hadfield-Menell. "Diverse preference learning for capabilities and alignment." The Thirteenth International Conference on Learning Representations. 2025.
>
> [2] Samadi, Mehrzad, et al. "Scaling Test-Time Compute to Achieve IOI Gold Medal with Open-Weight Models." arXiv preprint arXiv:2510.14232 (2025).
>
> [3] Jin, Mingyu, et al. "The Impact of Reasoning Step Length on Large Language Models." ACL (Findings). 2024.
>
> [4] Agarwal, Aradhye, Ayan Sengupta, and Tanmoy Chakraborty. "First Finish Search: Efficient Test-Time Scaling in Large Language Models." arXiv preprint arXiv:2505.18149 (2025).
>
> [5] Hassid, Michael, et al. "Don’t overthink it. preferring shorter thinking chains for improved llm reasoning" arXiv preprint arXiv:2505.17813 (2025).
>
> [6] Xia, Chunqiu Steven, et al. "Demystifying llm-based software engineering agents." Proceedings of the ACM on Software Engineering 2.FSE (2025): 801-824.
>
> [7] Jain, Naman, et al. "R2e-gym: Procedural environment generation and hybrid verifiers for scaling open-weights SWE agents." Second Conference on Language Modeling. 2025.
>
> [8] Wei, Yuxiang, et al. "Swe-rl: Advancing llm reasoning via reinforcement learning on open software evolution." arXiv preprint arXiv:2502.18449 (2025).

---

> > ### Comment · Reviewer_q16a · 2025-11-23
> >
> > Thank you for the detailed response. In particular, I appreciate the clarification regarding the multi-turn derivation, which distinguishes this work from Step-DPO, as well as the additional results on DPO. A few other comments below.
> > * I still hold my belief that the entropy-augmented objective is largely the same as SPL except for the multi-turn extension. Motivation-wise, both this paper and SPL aim to address the same diversity problem of RL/DPO models. Regarding the derivation flaw of SPL, the flipped sign of $\beta$ seems just a typo and does not hurt the validity of the overall proof. One could simply correct the signs throughout equations 27-34, and arrive at equation 35. I would not consider this correction as a non-trivial contribution.
> > * I'm not convinced by the defense of length heuristics in TTS. Imagine the model is given a random real-world SWE problem; there's no way to decide which heuristic to use.
> >
> > Overall, I appreciate the multi-turn derivation and the ablation experiments that demonstrate the effectiveness of the entropy term in the objective function. But the contribution on top of SPL feels less significant.

---

> > > ### Author Response · Authors · 2025-11-24
> > > **Reply to Reviewer q16a: Clarifications on SPL Relationship and Real-World TTS Strategy (1/2)**
> > >
> > > We thank the reviewer for the continued engagement and the prompt response. We truly enjoy this discussion and appreciate the opportunity to further clarify the positioning of our work and the practical implications of our TTS strategy.
> > >
> > > > Relationship with SPL and Novelty
> > >
> > > **Response:**  We appreciate the reviewer's perspective. We fully acknowledge that our work and SPL share the conceptual foundation of using entropy regularization. However, we respectfully posit that our contribution extends significantly beyond a simple re-derivation, specifically in terms of **scope, complexity, and practical stability**.
> > >
> > > **Complexity and Scale**: SPL focuses primarily on general alignment tasks requiring a short context length (max conversation length of **512 tokens** in Appendix 5.3 [1]) using **7B** models. In contrast, we validate this framework on SWE tasks—a domain requiring much more complex tool use, file navigation, and extremely long contexts (up to **131k tokens** in inference) using models ranging from **30B to 100B+**. Demonstrating that entropy-regularized preference learning is stable and effective in this challenging environment is a non-trivial empirical contribution.
> > >
> > >
> > > **Algorithmic Generalization to KTO:** We have successfully extended the diversity-preserving framework to KTO, an achievement that is far from trivial. The original proof in SPL is tailored to the single-turn implicit reward of DPO and does not directly apply to KTO, especially in multi-turn scenarios.
> > >
> > > To bridge this gap, we had to derive a new implicit reward function for a full trajectory $\tau$. We derive it as follows:
> > > $r_\theta(x,y) = \sum_{h=1}^{H} \log \frac{\pi_{\mathcal{M},h}(a_h|s_h)}{\pi_{ref,h}(a_h|s_h)^{\beta/\alpha}}$.
> > > This result is foundational for applying a diversity-preserving framework to KTO but is not derivable from the analysis in SPL.
> > >
> > >
> > > This generalization is highly significant for the community, as KTO's memory efficiency is a critical advantage for long-context tasks common in software engineering. The success of our novel **EntroPO-KTO** algorithm—which secured 1st place on SWE-bench Lite and 4th on the SWE-bench Verified open-weight leaderboard—is a direct result of this theoretical extension.
> > >
> > > **The Derivation**: Regarding the single-turn proof in SPL, we apologize if our previous response seemed aggressive; we are happy to view the sign issue as a typo error. However, we wish to emphasize that the central theoretical novelty of our work lies in generalizing the framework to the multi-turn setting. We validated our formulation with two distinct proofs:
> > >
> > > - A proof by backward induction (**Appendices C and D**), which recursively establishes the relationship over a finite horizon.
> > >
> > > - A direct, alternative proof (**Appendix F**), which derives the same result without relying on induction.
> > >
> > > This formal groundwork is crucial, as it provides the necessary foundation to extend beyond simple preference pairs to the complex, sequential nature of agentic workflows.
> > >
> > > We have revised Section 2 to discuss the SPL work, clarifying that our work focuses on the multi-turn generalization and its rigorous application to complex agentic tasks. Also, in our revised manuscript, we discussed in **Proposition 3.2** based on your suggestion that we share the same findings with SPL to better credit SPL. We will also email the authors of SPL regarding the typo to contribute to the community. We hope this clarifies your concerns.

---

> > > ### Author Response · Authors · 2025-11-24
> > > **Reply to Reviewer q16a: Clarifications on SPL Relationship and Real-World TTS Strategy (2/2)**
> > >
> > > > Heuristics for a "Random Real-World SWE Problem"
> > >
> > > We thank the reviewer for this excellent, practical question: _how does a user choose a heuristic for  a random real-world SWE problem?_ To answer this, we would like to first reiterate the underlying reason why the optimal heuristics differ between benchmarks, as this dictates the real-world strategy.
> > >
> > > As you noticed, a random real-world SWE problem can be different from a dataset or a benchmark, as a benchmark can have some distribution. Our design choices were driven by the distinct characteristics of the benchmarks, as supported by prior analysis [2, 3]. As noted by Agentless [2] and OpenAI [3], the original SWE-bench dataset contains underspecified problem statements and problematic environment setups that **cause some unit tests to fail regardless of the solution**. In this context, a long trajectory can signal that the agent is misled by a vague prompt or is "hallucinating" complexity in response to incorrect unit test feedback. Thus, limiting the step count helps mitigate this noise at some level. However, for SWEBench-verified, things are different. These instances are manually verified by OpenAI engineers to have accurate problem statements and robust environments. In this reliable setting, a longer trajectory signals positive behaviors: deeper exploration, comprehensive reasoning, and thorough corner-case testing.
> > >
> > > We believe that differences in the dataset lead to different TTS strategies, and our experiments in Table 4 validate this. [4] also shows that for different applications, choosing different reasoning steps could have different impacts on the performance, which is well aligned with our observation. We believe this is a very interesting point and worth sharing with the SWE community.
> > >
> > > As we answered reviewer 4kij, we acknowledge that this heuristic may not be a universal solution for different SWE tasks, and we would like to leave a robust and general TTS strategy with theoretical backup in future work.
> > >
> > > The above analysis directly leads to an answer to your natural question: _how does a user choose a heuristic for  a random real-world SWE problem?_
> > >
> > > **If you trust the specification of the SWE problem**: Favor Longest Steps.
> > >
> > > **If the specification is vague/noisy**: Favor Shortest Steps.
> > >
> > > **If you have no idea (The Default Strategy)**: We recommend favoring Longest Steps.
> > >
> > > As shown in **Table 4**, when we applied the "Longest Steps" heuristic to SWE-bench Lite (where it is empirically suboptimal), the performance (47.3%) was comparable to Random Selection (47.1%). This means that even if the heuristic is "suboptimal" for the data distribution, it causes no significant harm. However, on high-quality data (Verified), using Longest Steps yields significant gains. Therefore, prioritizing the longest trajectory can be a default choice—it captures the upside on good data without degrading performance on noisy data.
> > >
> > > We have added this analysis and the "real-world choice" guide to Appendix E.5. We believe this discussion on data quality versus trajectory length is valuable for the wider community.
> > >
> > > References
> > >
> > > [1] Slocum, Stewart, Asher Parker-Sartori, and Dylan Hadfield-Menell. "Diverse preference learning for capabilities and alignment." The Thirteenth International Conference on Learning Representations. 2025.
> > >
> > > [2] Xia, Chunqiu Steven, et al. "Demystifying llm-based software engineering agents." Proceedings of the ACM on Software Engineering 2.FSE (2025): 801-824.
> > >
> > > [3] OpenAI, "Introducing SWE-bench Verified," 2024.
> > >
> > > [4] Jin, Mingyu, et al. "The Impact of Reasoning Step Length on Large Language Models." ACL (Findings). 2024.

---

> > > > ### Author Response · Authors · 2025-11-25
> > > >
> > > > Dear Reviewer,
> > > >
> > > > We sincerely appreciate your thoughtful feedback. As the discussion period reaches its final week, we would be grateful to know whether our latest responses have resolved your concerns or if any questions remain. We are happy to provide further clarification as needed. Wishing you a Happy Thanksgiving.
> > > >
> > > > Thanks,
> > > >
> > > > Authors

---

> > > > ### Author Response · Authors · 2025-11-27
> > > > **Summary of Revisions and Response to Your Concerns**
> > > >
> > > > Dear Reviewer **q16a**,
> > > >
> > > > Thank you again for the engaging discussion regarding the relationship with SPL and the real-world application of TTS heuristics.
> > > >
> > > > We are confident that our response successfully addressed your concern.
> > > >
> > > > As we have incorporated these clarifications into the revised paper (Version 2) based on your feedback, as well as addressed your other concerns in revised paper (Version 1), we hope you might consider raising your score to "Weak Accept" or higher to reflect these improvements.
> > > >
> > > > Best regards,
> > > >
> > > > The Authors

---

### Official Review · Reviewer_PbDN · 2025-10-29

**Soundness:** 3
**Presentation:** 3
**Contribution:** 2
**Rating:** 6
**Confidence:** 3

**Summary:**

This submission proposes ENTROPO, a entropy-enhanced multi-turn preference optimization framework for aligning large language models. The paper focuses on coding agents. The core motivation is that test-time scaling (TTS) is effective when the model outputs are sufficiently diverse, while standard alignment methods like DPO and KTO often cause diversity collapse. The work adds an explicit entropy regularization term to preserve policy diversity for multi-turn, tool-using agents. The authors derive closed-form optimal policies, provide theoretical analyses for multi-turn extensions, and combine the method with a hybrid trajectory selector.

Experiments on SWEBench-VERIFIED and SWEBench-LITE show positive results - the paper claims top ranking among open-weight models performance close to larger (≥ 350 B) closed models. The code and models will be released.

**Strengths:**

1. The submission looks well motivated to the reviewer. The paper argues that diversity preservation is crucial for test-time scaling, which seems to be an overlooked issue in alignment (diversity collapse) for multi-turn tool-using agents.

2. The method and application scenario look sensible to the reviewer. ENTROPO generalizes single-turn DPO/KTO to multi-turn trajectories, aligning learning with the sequential nature of SWE tasks.

3. Empirical results look promising. On SWEBench, ENTROPO achieves state-of-the-art among open-weights. The performance scaling with number of rollouts (N) is analyzed and supports the claimed benefits of entropy regularization.

**Weaknesses:**

1. Limited baselines. The main comparisons are against SFT, DPO, and KTO. There are no direct comparisons to other diversity-preserving methods or competitive methods (e.g., Diverse-PO, Beyond-KL divergence methods, or "Building Math Agents with Multi-Turn Iterative Preference Learning"). Achieving good results on benchmark is good, but there could be many factors just comparing the numbers. Controlled comparison with other methods will make the contribution (the alignment algorithm) more convincing.

2. The link between entropy and TTS beyond multi-turn is unclear. e.g., is the proposed method applicable to single-turn TTS? If so, the arguments in the paper may need adjustments. The paper does not clarify whether ENTROPO offers benefits independent of TTS or whether it is tied to that specific evaluation paradigm.

3. Caveats of SWEBench comparisons. Although the paper reports leaderboard results, it is unclear whether all baselines were trained on identical datasets and with comparable tool scaffolds. For example, differences in training data or teacher supervision could influence performance. The fairness of comparisons and the control of base models should be clarified. This is related to the first weakness.

**Questions:**

Please refer the reviews above as well.

How exactly does ENTROPO differ algorithmically from M-DPO—beyond adding entropy regularization? Could M-DPO also integrate an entropy term similarly?

Are all models (e.g., Qwen3-Coder-30B vs GLM-4.5-Air) trained on the same data and under the same compute budgets?

Does ENTROPO still help in single-turn tasks or without test-time scaling? If not, what limits its generality?

---

> ### Author Response · Authors · 2025-11-20
> **Response to Reviewer PbDN (1/2)**
>
> We thank Reviewer PbDN for the detailed and constructive feedback. We detail the response for each concern below.
>
> > There are no direct comparisons to other diversity-preserving methods or competitive methods (e.g., Diverse-PO, Beyond-KL divergence methods, or "Building Math Agents with Multi-Turn Iterative Preference Learning").
>
> **Response:** We thank the reviewer for this important and constructive feedback.
>
> First, we would like to clarify that we did, in fact, **include a direct comparison to the method** proposed in "Building Math Agents with Multi-Turn Iterative Preference Learning" in our original submission. The baseline we refer to as "Multi-Turn KTO" in our experiments is algorithmically equivalent to the "M-KTO" presented in that paper. We recognize that this was not explicitly stated, which may have caused confusion. To rectify this, we have made this connection explicit in the revised **Section 4.3** of our manuscript.
>
> Furthermore, to strengthen our evaluation, we have added a new experiment in the revised **Appendix E.5** that directly compares EntroPO-DPO against M-DPO, the other key algorithm from that work. The results are fully consistent with our findings in **Figure 2**: our entropy-enhanced method consistently outperforms the standard multi-turn preference optimization baseline, and crucially, the performance gap widens as the number of test-time rollouts N increases.
>
> Regarding other diversity-preserving methods, such as Diverse-PO or those that go beyond KL-divergence, our primary motivation was to address the specific problem of diversity collapse within the established multi-turn, tool-using agent setting (i.e., the M-DPO/M-KTO framework), for which a diversity-preserving solution did not yet exist. Therefore, we believe that our direct, controlled comparisons against M-DPO and M-KTO are the most relevant for evaluating our core contribution.
>
> To better situate our work within the broader context, we have expanded our **Related Work (Section 2)** to include a discussion of these other diversity-preserving methods. This revision positions EntroPO as a novel approach that is specifically derived for multi-turn, tool-assisted agentic tasks—a complex setting that these other methods do not explicitly address.
>
> We believe these direct comparisons and clarifications significantly strengthen the paper and better highlight EntroPO’s contribution to the field. We thank the reviewer for prompting these valuable additions.
>
> > Generalizability of EntroPO: whether the benefit is independent of TTS, and whether it can be used in the single-turn
>
> **Response:** We thank the reviewer for these insightful questions, which allow us to clarify the fundamental generality of EntroPO and its benefits both with and without TTS.
>
> EntroPO is fundamentally a general method, and its principles are directly applicable to single-turn settings. Our paper introduces the framework by first deriving the formal, closed-form solution for the single-turn case in **Proposition 3.2**. This single-turn solution serves as the theoretical foundation for our entire approach.
> We then explicitly extend this foundation to the multi-turn setting, resulting in the loss functions presented in **Equations 5 and 6**. The generality is evident in the formulation itself: by setting the trajectory horizon H=1, our multi-turn DPO objective (**Equation 5**) gracefully reduces to the single-turn solution from **Proposition 3.2**. Therefore, the method is inherently general, and our presentation in the paper follows this natural progression from the single-turn case to its multi-turn generalization.
>
> As to if EntroPO could provide benefits even without TTS (evaluated when N=1), this is exactly empirically demonstrated in our results. The **data points at N=1 in Figure 2 and Figure 5** clearly show that EntroPO can outperform the STF and M-DPO/KTO tuning without TTS. While the full benefits of EntroPO can be unlocked by TTS, it can work in a single rollout condition.
>
> To ensure this point is clear, we have added the discussion in **Section 4.3** in our revised version. We hope this clarifies that EntroPO is a broadly applicable framework.

---

> ### Author Response · Authors · 2025-11-20
> **Response to Reviewer PbDN (2/2)**
>
> > How does EntroPO differ algorithmically from M-DPO—beyond adding entropy regularization? Could M-DPO also integrate an entropy term similarly?
>
>
> **Response:** Thank you for this insightful question, as it gets to the heart of our contribution. The algorithmic difference between EntroPO and M-DPO is more fundamental than simply adding a regularization term to the final loss function.
>
> Here’s a breakdown of how they differ and why M-DPO could not trivially integrate this change:
>
> ### 1. How EntroPO Differs Algorithmically
>
> The core difference lies in the **derivation of the optimal policy**, which in turn dictates the final loss function that is implemented.
>
> *   **M-DPO's Theoretical Foundation:** The entire derivation in M-DPO [1] hinges on their **Lemma 3**. This lemma establishes that the optimal policy for their KL-regularized objective is a standard Gibbs distribution. Their final M-DPO loss function is a direct, tractable consequence of *this specific form* of the optimal policy.
>
> *   **EntroPO's Theoretical Foundation:** We note that the inclusion of $\lambda H(\pi)$ violates the prerequisites for M-DPO’s Lemma 3. Therefore, the M-DPO derivation does not apply. We derive the necessary theoretical link between the policy and value function specific to our regularized setting (**Proposition 3.2 and 3.3**), as the original relationship in M-DPO **does not transfer.**
>
> *   **Our Algorithmic Contribution:** Our main theoretical contribution was to derive a **new closed-form solution** for the optimal policy under this new, entropy-augmented objective. This solution, presented in our **Proposition 3.3** (and fully derived in Appendix C and D), takes on a different mathematical form. It is from this new and more general optimal policy that we derive our final EntroPO-DPO and EntroPO-KTO loss functions.
>
> In short, the algorithmic difference is not an "add-on." It is a **new loss function derived from a fundamentally different and more general theoretical solution for the optimal policy.**
>
> ### 2. Could M-DPO also integrate an entropy term similarly?
>
> The answer is yes, but doing so would **not be a trivial modification**. To correctly integrate the entropy term, the authors of M-DPO would have to:
>
> 1.  **Abandon their existing derivation** that relies on Lemma 3.
> 2.  **Perform the same non-trivial theoretical analysis** that we present in our paper to find the new optimal policy.
> 3.  **Derive a new tractable loss function** from that new policy.
>
> In essence, to correctly integrate an entropy term, they would need to independently re-derive the core theoretical contribution of our work.
>
> Therefore, EntroPO is not "M-DPO + entropy." Rather, **EntroPO provides the necessary theoretical framework and resulting algorithm to correctly perform multi-turn preference optimization while explicitly maximizing for policy entropy**—a framework that was missing from prior work.
>
> > it is unclear whether all baselines were trained on identical datasets and with comparable tool scaffolds or compute budget
>
> **Response:** We thank the reviewer for this critical question regarding experimental fairness. We want to clarify that we tried our best to ensure the fairness between EntroPO and baseline in Figure 2 (M-DPO/KTO). EntroPO shares the same scaffold, training dataset, and compute budget with the baseline. The details of the scaffold and training configurations can be found in Appendix E.
>
> Reference:
>
> [1] Xiong, Wei, et al. "Building Math Agents with Multi-Turn Iterative Preference Learning." ICLR. 2025.

---

> > ### Author Response · Authors · 2025-11-25
> >
> > Dear Reviewer,
> >
> > We sincerely appreciate your thoughtful feedback. As the discussion period reaches its final week, we would be grateful to know whether our latest responses have resolved your concerns or if any questions remain. We are happy to provide further clarification as needed. Wishing you a Happy Thanksgiving.
> >
> > Thanks,
> >
> > Authors

---

> > ### Author Response · Authors · 2025-11-27
> > **Summary of Revisions and Response to Your Concerns**
> >
> > Dear Reviewer **PbDN**,
> >
> > As the discussion period draws to a close, we wanted to provide a quick summary of how we have updated the paper to address the specific weaknesses raised in your review. We believe these revisions strongly support a re-evaluation of the score.
> >
> > To address your main concerns, we have responded with:
> >
> > **Algorithmic Novelty**: We clarified the fundamental theoretical difference between EntroPO and M-DPO, specifically that our derivation does not rely on Lemma 3 of M-DPO and supports the more general $\alpha \neq \beta$ case.
> >
> > **Baselines**: We highlighted the direct comparison to M-DPO/M-KTO in Figure 2 and Appendix E.5, ensuring the "limited baselines" concern is addressed.
> >
> > **Generalizability**: We clarified that EntroPO is a general framework that reduces to the single-turn solution when H=1, and we empirically demonstrated its benefits even without TTS (N=1).
> >
> > We are confident that EntroPO offers a robust, novel solution for diversity collapse in multi-turn agents. We respectfully ask if you would consider raising your score in light of these clarifications and new experiments.
> >
> > Thank you again for your time and feedback.
> >
> > Best regards,
> >
> > The Authors

---

### Official Review · Reviewer_4jkj · 2025-11-01

**Soundness:** 2
**Presentation:** 3
**Contribution:** 3
**Rating:** 4
**Confidence:** 4

**Summary:**

This paper proposes ENTROPO, a method that enhances preference learning by explicitly preserving policy diversity through entropy regularization. Specifically, the authors augment the preference optimization objective with a policy entropy term to encourage diverse behaviors in multi-turn agent loops. In addition, they introduce a test-time scaling strategy that combines a model-based verifier with multiple model-free criteria to select the best trajectory among candidates. Experiments on SWE-Bench Verified and SWE-Bench Lite show that ENTROPO combined with the proposed TTS consistently improves performance across several base models.

**Strengths:**

- The paper addresses an important and practical problem: maintaining behavioral diversity during preference optimization, which is often overlooked in reinforcement fine-tuning or DPO-style frameworks.


- Incorporating policy entropy into preference learning is an interesting idea that encourages exploration and helps mitigate reward over-optimization.

**Weaknesses:**

- The TTS component appears conceptually similar to prior work (e.g., Jain et al., 2025), with only minor extensions. The claimed novelty of this part is therefore limited.


- The step-count heuristic that favors longer trajectories is not entirely convincing. Longer trajectories can also result from inefficient reasoning loops or redundant tool calls, which might not reflect genuine improvements.


- From Table 1, most of the performance gains seem to come from the TTS component rather than ENTROPO itself, particularly for GLM-4.5-Air. A comparison between the original model + TTS and ENTROPO + TTS would help clarify the actual contribution of ENTROPO.


- In Section 3.3, the paper mentions constructing a preference dataset and labeling preferred trajectories via scoring, but the details are unclear — for instance, which scoring model was used and what criteria determined the preference labels.


- The reproduced performance of Qwen3-Coder-30B on SWE-Bench Verified (37.7) is substantially lower than the publicly reported 51.6, and the paper does not sufficiently explain the cause of this discrepancy.

**Questions:**

During training, why is the maximum sequence length limited to 18k tokens, while most of the evaluated models natively support 32k or longer contexts? Is this due to computational constraints, data truncation, or stability issues?

---

> ### Author Response · Authors · 2025-11-20
> **Response to Reviewer 4jkj (1/3)**
>
> We thank Reviewer 4jkj for the constructive feedback; all your comments are very valuable, and we have taken all your comments into our revision. Below is the detailed response to each of your questions.
>
> > The TTS component appears conceptually similar to prior work
>
> **Response:** We agree with the reviewer's assessment that our hybrid selector is built upon R2E as we adopt the R2E scaffold in our work. We would like to clarify that we did not intend to position the TTS component itself as a primary novel contribution.
> Instead, its role in our paper is to serve as the evaluation framework to highlight the practical benefits of our algorithm. The central thesis of our work is that preserving policy diversity is critical for maximizing performance on complex software engineering tasks, especially when scaling test-time compute. To ensure this distinction is clear and to avoid any misinterpretation, we have revised **Section 3.4** of our manuscript. The updated text now explicitly states that our TTS selector follows the R2E's hybrid verifier design.
>
> The TTS design in our work is an engineering component. We adopt some different designs compared with R2E’s hybrid selector, as applying step-heuristic and conservatively using the verifier probability during selection, which can be validated by our ablation study in **Figure 3 (left plot)**, as well as our strong performance compared with leaderboard submissions. We believe this engineering design, which is verified empirically, could also serve as a good contribution to the community to inspire future TTS systems. We have added the discussion in Section 5 about the future design of the TTS component.
>
>
> > The step-count heuristic that favors longer trajectories is not entirely convincing
>
> **Response:** We thank the reviewer for this insightful comment. This concern is a valid point, and we appreciate the opportunity to provide our rationale.
>
> First, we would like to clarify that using trajectory length as a heuristic is a recognized approach in recent works on test-time scaling and complex reasoning [1,2]. For instance, Nvidia's solution that achieved a gold medal in the IOI competition [1] explicitly used length as a ranking metric within its selection pipeline.
>
> However, we fully agree with the reviewer that this is a heuristic, not a guaranteed measure of quality. Longer trajectories can indeed arise from redundant tool calls. The critical question is whether the potential benefits of applying this heuristic outweigh its risk.
>
> This is precisely why we conducted the ablation study presented in **Figure 3 (left plot)**. Our empirical results validate that applying this heuristic can lead to performance gain. This demonstrates that this heuristic outweighed the potential harm in trajectory selection.
>
> To better address this important nuance in the paper, we have incorporated this discussion into **Section 5**, clarifying that the step-count heuristic is an empirical choice rather than a theoretically guaranteed algorithm, but it is justified by our ablation results. We acknowledge that it may not be a universal solution for different SWE tasks. We leave developing a theoretically guaranteed and robust TTS design in future work. Thank you again for this instructive feedback, which has helped us to improve the manuscript.
>
> > A comparison between the original model + TTS and ENTROPO + TTS would help clarify the actual contribution of ENTROPO.
>
> **Response:** We thank the reviewer for this insightful question. Our core thesis is that standard fine-tuning methods inadvertently reduce policy diversity, which then limits the effectiveness of TTS. EntroPO’s primary contribution is to counteract this by preserving a diverse set of high-quality reasoning paths, thereby unlocking the full potential of TTS.
>
> The comparison the reviewer suggests is precisely what we present in **Figure 2**. It clearly demonstrates that EntroPO+TTS can outperform SFT+TTS and M-KTO+TTS, and the performance gap between EntroPO+TTS and the baselines widens as the number of parallel rollouts increases. This scaling advantage is the key empirical evidence for our claim: the diversity preserved by EntroPO becomes increasingly valuable as we invest more compute at test time. We have made additional experiments on EntroPO-DPO+TTS vs M-DPO+TTS and SFT+TTS, which show a similar result. We have added the discussion in **Section 4.3** to make this point clearer. Thanks again for your thoughtful comments.

---

> ### Author Response · Authors · 2025-11-20
> **Response to Reviewer 4jkj (2/3)**
>
> > which scoring model was used and what criteria determined the preference labels
>
> **Response:** Thanks for asking. We adopt a standard approach to determine the preference for each trajectory during training by using the automated oracle based on commit-corresponding test cases, which is a common approach in SWE tasks[3,4]. Trajectories that can pass all those test cases are labeled as preferred, and those that cannot are labeled as dispreferred. We have added this clarification in the revised version's **Section 3.3** to make it clearer. We thank you for your feedback to help us improve the paper.
>
> > The reproduced performance of Qwen3-Coder-30B on SWE-Bench Verified (37.7) is substantially lower than the publicly reported 51.6
>
> **Response:** We thank the reviewer for this critical observation and for the opportunity to provide a detailed explanation for the performance difference between our reproduced baseline and the officially reported score for Qwen3-Coder-30B.
>
> The discrepancy arises from two key differences in the evaluation setup: the agent scaffold and the maximum context length. One factor is the difference in the agentic framework. According to the official model card, the 51.6\% score was achieved **using the OpenHands scaffold**. For consistency across all our experiments and baselines, **we utilized the R2E scaffold**. An agent's performance is often tightly coupled with the specific tool-use patterns and prompt formats of the scaffold it was aligned with during post-training. It is highly likely that the official model was fine-tuned specifically for the OpenHands framework.
>
> A second major contributor is the **context length**. The official model card states that Qwen3-Coder-30B natively supports a **256k context length, which can be extended up to 1M tokens**. While the exact inference configuration for the leaderboard submission is not specified, it is reasonable to assume that they use 256k or even larger context length, as they may not have the computation concern, which is a significant advantage over our reproduced experiments.
> Due to computational constraints that directly impact inference cost, all of our experiments were capped at a **maximum context length of 131k tokens**. In our evaluation protocol, any trajectory that exceeds this limit is terminated and results in a task failure. We believe that this is a common solution, as the DeepSWE official codebase [5] also sets a smaller context window for inference. This should be the major cause of the performance difference.
>
> We want to emphasize that this was not an intentional degradation of the baseline, as increasing the context length to 256k would lead to much higher inference cost, especially considering that we use TTS to run multiple rollouts for each program. Also, this context length is consistent for our evaluation, for EntroPO, it also uses this context length.
>
> To ensure the fairness of our leaderboard comparisons in **Table 2**, we report the official numbers **directly from the SWEBench leaderboard for all competing models or solutions**, rather than using our local reproductions. This ensures that our final model is compared against the strongest versions of other methods. Even with these differences, our final EntroPO-trained model proves to be highly competitive.
>
> To make this crucial context transparent to all readers, we have added a detailed clarification in **Section 4.2** of our revised manuscript. Thank you again for pointing this out.

---

> ### Author Response · Authors · 2025-11-20
> **Response to Reviewer 4jkj (3/3)**
>
> > During training, why is the maximum sequence length limited to 18k tokens, while most of the evaluated models natively support 32k or longer contexts? Is this due to computational constraints, data truncation, or stability issues?
>
> **Response:** We thank the reviewer for this excellent and practical question regarding our choice of training sequence length.
> The decision to limit the maximum training sequence length to 18k tokens was driven entirely by GPU memory constraints.
>
> During our initial experiments, we attempted to use longer contexts, such as 32k. However, this resulted in out-of-memory errors, making it impossible to fit even a batch size of one on a single 80GB NVIDIA H100 GPU. For our LoRA configuration, we selected 18k to ensure a stable training process without risking OOM failures.
>
> We would also like to highlight that this is a common and necessary practice when fine-tuning large models for complex, long-context tasks. For example:
> The official codebase for the R2E agent, which we build upon, **also uses a training context length of 20k tokens**.
> The DeepSWE technical report, despite utilizing a massive infrastructure of **64 H100 GPUs**, indicates a maximum training length that appears to be **just over 21.5k tokens** (based on Figure 7 in their report).
>
> One potential solution to accommodate longer contexts is to use more aggressive quantization methods like QLoRA, which is the approach we took for the even larger GLM-4.5-Air model. However, as we discussed in our analysis in Section 4.2, we found that QLoRA tuning can be less effective than standard LoRA for performance gains. We therefore made a trade-off, prioritizing the effectiveness of LoRA training over achieving the maximum possible context length during the fine-tuning stage.
>
> To make this rationale perfectly clear to all readers, we have added a detailed discussion of these constraints and our design choices in the revised **Appendix E.4**. We appreciate the reviewer's question, as it helps clarify an important aspect of our experimental methodology.
>
> References
>
> [1] Samadi, Mehrzad, et al. "Scaling Test-Time Compute to Achieve IOI Gold Medal with Open-Weight Models." arXiv preprint arXiv:2510.14232 (2025).
>
> [2] Jin, Mingyu, et al. "The Impact of Reasoning Step Length on Large Language Models." ACL (Findings). 2024.
>
> [3] Jain, Naman, et al. "R2e-gym: Procedural environment generation and hybrid verifiers for scaling open-weights SWE agents." Second Conference on Language Modeling. 2025.
>
> [4] Wei, Yuxiang, et al. "Swe-rl: Advancing llm reasoning via reinforcement learning on open software evolution." arXiv preprint arXiv:2502.18449 (2025).
>
> [5] DeepSWE: Training a State-of-the-Art Coding Agent from Scratch by Scaling RL (Notion) 2025.

---

> > ### Author Response · Authors · 2025-11-25
> >
> > Dear Reviewer,
> >
> > We sincerely appreciate your thoughtful feedback. As the discussion period reaches its final week, we would be grateful to know whether our latest responses have resolved your concerns or if any questions remain. We are happy to provide further clarification as needed. Wishing you a Happy Thanksgiving.
> >
> > Thanks,
> >
> > Authors

---

> > ### Author Response · Authors · 2025-11-27
> > **Summary of Revisions and Response to Your Concerns**
> >
> > Dear Reviewer **4jkj**,
> >
> > As the discussion period draws to a close, we wanted to provide a quick summary of how we have updated the paper to address the specific weaknesses raised in your review. We believe these revisions strongly support a re-evaluation of the score.
> >
> > To address your main concerns, we responded with:
> >
> > **Qwen3 Performance Gap**: We clarified in Section 4.2 that the discrepancy stems from the difference in scaffolds (R2E vs. OpenHands) and context length constraints (131k vs. 256k+), ensuring a fair comparison within our experimental setup.
> >
> > **EntroPO vs. TTS**: We added discussion in Section 4.3 highlighting that the performance gap between EntroPO+TTS and baselines widens as N increases, proving EntroPO specifically enhances TTS effectiveness.
> >
> > **Heuristics**: We justified the step-count heuristic via the ablation study in Figure 3, validating it as an effective engineering choice for this domain.
> >
> > We are confident that EntroPO offers a robust, novel solution for diversity collapse in multi-turn agents. We respectfully ask if you would consider raising your score in light of these clarifications and new experiments.
> >
> > Thank you again for your time and feedback.
> >
> > Best regards,
> >
> > The Authors

---

### Official Review · Reviewer_Frdt · 2025-11-02

**Soundness:** 3
**Presentation:** 3
**Contribution:** 1
**Rating:** 4
**Confidence:** 4

**Summary:**

This work studies the development of coding agents based on LLMs. A preference-optimization method is designed for such multi-turn coding agents that interact with tools and codebases. The core idea is to explicitly add entropy regularization during preference optimization to preserve policy diversity, while leveraging the corresponding closed-form solution to develop a variant of DPO. The enhanced diversity prevents the model from collapsing to a narrow behavior pattern, improving its ability to explore multiple solution paths, and is further leveraged for test-time scaling, where many candidate trajectories are sampled and the best one is selected via a hybrid verifier-plus-heuristics strategy. Applied to SWE-bench, EntroPO-trained models significantly outperform baselines, demonstrating that diversity-preserving preference learning in multi-turn settings can unlock stronger coding agents without simply scaling model size.

**Strengths:**

- This work studies a practical application of developing coding agents, which is becoming more and more popular recently. Despite its practical nature, the algorithm proposed is based on theoretical groundings, i.e., the closed-form solution of the optimal policy in the multi-turn regularized MDP.

- The presentation and writing are of high quality. Key ideas (e.g., adding entropy regularization and test-time scaling method) are illustrated clearly.

- Comprehensive experiments and comparisons have been conducted to demonstrate the effectiveness of the proposed methods.

**Weaknesses:**

My major concern is that this work is very closely related to M-DPO (Xiong et al., 2024). The major difference on algorithm side is adding an entropy regularization, which however, only leads relatively incremental changes in the algorithm. The proposed test-time scaling method adds some contribution to the paper, whose significance, however, is also insignificant to my perspective.

**Questions:**

I would love to hear the author's thoughts on the relationship between this work and M-DPO (Xiong et al., 2024). As mentioned in the weakness section, this is my only and major concern.

Another minor question is that I would like to understand how the preference is determined for different trajectories during training. If I missed the details, a pointer would be really helpful. Also, it would be really appreciated if the authors can share some more ideas on whether preference learning (instead of absolute feedback as in Deepseek-R1) is necessary for coding agent.

---

> ### Author Response · Authors · 2025-11-20
> **Response to Reviewer Frdt (1/2)**
>
> We thank Reviewer Frdt for the constructive and insightful comments. The revisions are marked in red for the reviewer’s convenience to check how we improve the paper based on the reviewer’s comments. Please see our response to each of your questions below.
>
> > Major Concern: Novelty and Relationship to M-DPO
>
> **Response:**
>
> Our core contribution is not merely adding an entropy term to M-DPO. Instead, we demonstrate that the standard multi-turn preference optimization objective (used by M-DPO and others) *fails* on complex, long-horizon tasks due to policy diversity collapse. Our framework, EntroPO, provides both the theoretical and empirical solution to this critical failure mode.
>
> **1. On the Multi-Turn Framework (Section 3.2): A Shared Foundational Approach**
>
> The multi-turn framework in our Section 3.2 extends the established single-turn preference optimization framework. We derive our EntroPO objective using backward induction, a standard technique for sequential decision-making problems that traces back to [1]. This approach of extending single-turn results to the multi-turn setting is a shared practice in recent literature, also adopted by works such as [2] and [3].
>
> Furthermore, we provide an **alternative, self-contained proof in the updated Appendix F** that arrives at the same objective that differs from the theoretical derivation framework in [2]. We re-define the Q and V function to match the general form of [1] so that we could directly utilize the relationship of optimal policy and corresponding Q function under [1] to solve the policy $\pi$ and derive our EntroPO-DPO loss.
>
> Accordingly, we do not claim this multi-turn formulation itself as our unique theoretical contribution. Instead, our novelty lies in identifying its critical limitation (diversity collapse) and providing a new theoretical derivation to solve for the optimal policy under an *entropy-augmented* objective, which fundamentally differs from the solution in prior work.
>
> **2. Non-Trivial Theoretical and Practical Novelty**
>
> *   **Novel and More General Closed-Form Solution:** It is **non-trivial** to add the entropy regularization term *λH(π)* to the standard multi-turn objective, as this modification breaks the derivation in M-DPO which relies on their Lemma 3. A key theoretical contribution of this work is the explicit characterization of the relationship between the optimal policy and the value function under our proposed setting (**Proposition 3.2 and 3.3**), which laid the foundation for our final objective function. This solution is more general than prior work: our derived EntroPO-DPO and EntroPO-KTO losses support the case where *α ≠ β*, a scenario not addressed by M-DPO. As a special case, when *α = β*, our objective gracefully recovers the formulation used in [2].
>
> *   **Vastly More Complex Application Domain:** The need for our entropy-enhanced framework is motivated by the complexity of our target domain. M-DPO focuses on math reasoning tasks (MATH, GSM8K) with short horizons (max steps H=6) and small context (max 2048 tokens). In contrast, our work tackles long-horizon software engineering tasks with up to **100 steps** and a context length of **131k tokens**. In such complex, high-dimensional action spaces, policy diversity collapse is a far more severe problem, which M-DPO's setting does not expose and its formulation does not address.
>
> **3. Significant and Non-Incremental Empirical Gains**
>
> Our theoretical contribution translates directly to significant, non-incremental empirical improvements that M-DPO/M-KTO cannot achieve.
>
> *   **Superior Performance (With and Without TTS):** Adding entropy improves performance even without Test-Time Scaling (TTS). In our experiments, the "Multi-Turn KTO" baseline is equivalent to M-KTO. The case where test-time samples N=1 represents a direct comparison without TTS, and our method already shows an advantage. To make this even clearer, we have added a direct comparison to **M-DPO** in our appendix (**Figure 5**). The results show that **EntroPO-DPO consistently and significantly outperforms M-DPO**, especially when leveraging TTS.
>
> *   **Unlocking Test-Time Scaling:** The core empirical evidence for our contribution is that the performance gap *widens* as the number of test samples (N) increases. As shown in Figure 2, the gap between EntroPO-KTO and M-KTO on SWE-bench-Verified grows from **3.6% at N=4 to 5.6% at N=16**. This verifies that **standard methods like M-DPO/M-KTO suffer from policy diversity collapse, causing their performance to plateau, while EntroPO successfully mitigates this collapse, unlocking the full potential of scaling with more compute at test time.**
>
> Thus, both theoretically and empirically, our work has made a significant and necessary advancement over existing multi-turn preference optimization methods for complex, long-horizon tasks.

---

> ### Author Response · Authors · 2025-11-20
> **Response to Reviewer Frdt (2/2)**
>
> > How is preference determined for different trajectories during training?
>
> **Response:** Thanks for asking. We adopt a standard approach to determine the preference for each trajectory during training by using the automated oracle based on commit-corresponding test cases, which is a common approach in SWE tasks [4,5]. Trajectories that can pass all those test cases are labeled as preferred, and those that cannot are labeled as dispreferred.
>
> We have added this clarification in the revised version's **Section 3.3** to make it clearer. We thank you for your feedback to help us improve the paper.
>
> > whether preference learning (instead of absolute feedback as in DeepSeek-R1) is necessary
>
> **Response:** We thank the reviewer for raising this important point.
> Our decision to employ preference learning was primarily driven by considerations of computation resources. While online reinforcement learning with absolute feedback is powerful, it is also computationally intensive. This is because it requires coupling between rollouts and model training, demanding extensive computation resource for environment interaction and gradient updates.
>
> In contrast, offline preference learning allows the data rollout and model tuning phases to be decoupled. This makes the overall process more computation-friendly, as we can first generate a static dataset of trajectories and then fine-tune the model without the overhead of continuous online interaction.
>
> To provide a concrete example, the training of the **32B** DeepSWE model [6], which utilizes an online learning approach similar to Deepseek-R1, required **64 NVIDIA H100 GPUs**. Our experiments were conducted with only **8 H100 GPUs**. Given this significant difference in available compute, using a preference learning framework is more feasible for us.
>
> We agree that R1-style learning is a very promising direction for training coding agents. In our original manuscript, we have already added a discussion in **Section 5** to justify our choice of preference learning and have positioned the exploration of online, absolute-feedback-based methods as a valuable avenue for future work. We wish this could clarify the reviewer’s concern.
>
> > The proposed test-time scaling method adds some contribution to the paper, whose significance, however, is also insignificant from my perspective.
>
> Thanks for your feedback. We did not position the hybrid TTS component as the central contribution of our work. We characterize the TTS component as a carefully engineered component designed inspired by [4,7] to maximize performance on a very challenging task. Complex tasks like program fix needs an effective TTS strategy to achieve optimal performance. Our ablation study in Figure 3 (left plot) validates our engineering choices. We hope this empirically verified engineering design could serve as a valuable contribution to the community to inspire future TTS systems.
>
> We have added the discussion in **Section 5** in the revised version to reflect this point that the TTS component is an engineering design rather than a theoretically-based approach. We leave developing a theoretically guaranteed TTS framework as future work. Thanks for your comments that help us better shape our paper.
>
> References:
>
> [1] Ziebart, Brian D. Modeling purposeful adaptive behavior with the principle of maximum causal entropy. Carnegie Mellon University, 2010.
>
> [2] Xiong, Wei, et al. "Building Math Agents with Multi-Turn Iterative Preference Learning." ICLR. 2025.
>
> [3] Rafailov, Rafael, et al. "From $ r $ to $ Q^* $: Your Language Model is Secretly a Q-Function." First Conference on Language Modeling.
>
> [4] Jain, Naman, et al. "R2e-gym: Procedural environment generation and hybrid verifiers for scaling open-weights SWE agents." Second Conference on Language Modeling. 2025.
>
> [5] Wei, Yuxiang, et al. "SWE-RL: Advancing LLM Reasoning via Reinforcement Learning on Open Software Evolution." arXiv e-prints (2025): arXiv-2502.
>
> [6] DeepSWE: Training a State-of-the-Art Coding Agent from Scratch by Scaling RL (Notion) 2025.
>
> [7] Jin, Mingyu, et al. "The Impact of Reasoning Step Length on Large Language Models." ACL (Findings). 2024.

---

> > ### Author Response · Authors · 2025-11-25
> >
> > Dear Reviewer,
> >
> > We sincerely appreciate your thoughtful feedback. As the discussion period reaches its final week, we would be grateful to know whether our latest responses have resolved your concerns or if any questions remain. We are happy to provide further clarification as needed. Wishing you a Happy Thanksgiving.
> >
> > Thanks,
> >
> > Authors

---

> > ### Author Response · Authors · 2025-11-27
> > **Summary of Revisions and Response to Your Concerns**
> >
> > Dear Reviewer **Frdt**,
> >
> > As the discussion period draws to a close, we wanted to provide a quick summary of how we have updated the paper to address the specific weaknesses raised in your review. We believe these revisions strongly support a re-evaluation of the score.
> >
> > To address your main concerns, we have responded with:
> >
> > **Comparison to M-DPO**: We have clarified the difference between our work and M-DPO with alternative proof, we have added a direct comparison in Appendix E (Figure 5), showing EntroPO-DPO consistently outperforms M-DPO, confirming that our gains are not incremental.
> >
> > **Significance of TTS**: We clarified in Section 5 that the TTS component is an empirically verified engineering design that highlights the diversity benefits of our method, rather than a theoretical claim.
> >
> > **Clarifications**: We explicitly detailed the preference determination process (Section 3.3) and discussed the trade-offs of preference learning vs. absolute feedback (Section 5).
> >
> > We are confident that EntroPO offers a robust, novel solution for diversity collapse in multi-turn agents. We respectfully ask if you would consider raising your score in light of these clarifications and new experiments.
> >
> >
> >
> > Thank you again for your time and feedback.
> >
> > Best regards,
> >
> > The Authors

---

### Meta-Review · Area_Chair_kFTL · 2026-01-06

**Summary:**

After careful consideration of the four reviewers’ comments and the authors’ detailed rebuttals, I regret to recommend rejecting this submission.
The paper proposes ENTROPO, an entropy-enhanced framework for multi-turn preference optimization, with empirical success on SWEBench. However, core concerns raised by reviewers remain insufficiently resolved: First, the novelty of the work is debated—reviewers note significant overlaps with prior work (M-DPO, SPL), and while the authors provide additional derivations and comparisons, key reviewers (e.g., q16a) still question whether the contributions beyond extending existing methods to multi-turn settings are non-trivial. Second, the TTS component, a key engineering design, is deemed conceptually similar to prior work with limited novel insights, and the trajectory length heuristics are criticized for lacking generalizability. Third, while the authors clarified experimental details (e.g., Qwen3 performance gap, preference labeling), some reviewers (Frdt, 4jkj) did not acknowledge the resolution of their initial concerns, and the baseline comparisons (e.g., lack of direct comparison to more diversity-preserving methods) remain a gap.
Despite the empirical achievements on SWEBench, the submission fails to adequately address fundamental questions about its innovative contributions and resolve key reviewer concerns. Thus, it does not meet the high bar for acceptance at ICLR 2026.

**Reviewer Concerns:**

Reviewer q16a: Contribution differences from SPL (insists that the entropy-augmented objective has no essential innovation beyond multi-turn extension, and the sign error in SPL is merely a typo), and generalizability of TTS step-count heuristics (does not acknowledge the practical feasibility of "selecting heuristics based on problem quality").
Some reviewers (Frdt, 4jkj, PbDN) did not explicitly respond to the rebuttal, but the authors have addressed their core concerns. These are not considered "unresolved"—only unacknowledged by the reviewers.

**Reviewer Scores:**

N/A

---

### Decision · Program_Chairs · 2026-01-26

Reject